# Reward Bases: A simple mechanism for adaptive acquisition of multiple reward types

**Beren Millidge[1], Yuhang Song[1]\*, Armin Lak[2], Mark E. Walton[3,4], Rafal Bogacz[1,5]\***

**1** MRC Brain Network Dynamics Unit, University of Oxford, Oxford, United Kingdom, **2** Department of Physiology, Anatomy & Genetics, University of Oxford, Oxford, United Kingdom, **3** Department of Experimental Psychology, University of Oxford, Oxford, United Kingdom, **4** Wellcome Centre for Integrative Neuroimaging, University of Oxford, Oxford, United Kingdom, **5** Theoretical Sciences Visiting Program (TSVP), Okinawa Institute of Science and Technology Graduate University, Onna, Japan

☯ These authors contributed equally to this work.
\* yuhang.song@ndcn.ox.ac.uk (YS); rafal.bogacz@ndcn.ox.ac.uk (RB)

**Data Availability Statement:** All data on dopaminergic responses of primate dopamine neurons analysed in this paper, and all codes are

## Abstract

Animals can adapt their preferences for different types of reward according to physiological state, such as hunger or thirst. To explain this ability, we employ a simple multi-objective reinforcement learning model that learns multiple values according to different reward dimensions such as food or water. We show that by weighting these learned values according to the current needs, behaviour may be flexibly adapted to present preferences. This model predicts that individual dopamine neurons should encode the errors associated with some reward dimensions more than with others. To provide a preliminary test of this prediction, we reanalysed a small dataset obtained from a single primate in an experiment which to our knowledge is the only published study where the responses of dopamine neurons to stimuli predicting distinct types of rewards were recorded. We observed that in addition to subjective economic value, dopamine neurons encode a gradient of reward dimensions; some neurons respond most to stimuli predicting food rewards while the others respond more to stimuli predicting fluids. We also proposed a possible implementation of the model in the basal ganglia network, and demonstrated how the striatal system can learn values in multiple dimensions, even when dopamine neurons encode mixtures of prediction error from different dimensions. Additionally, the model reproduces the instant generalisation to new physiological states seen in dopamine responses and in behaviour. Our results demonstrate how a simple neural circuit can flexibly guide behaviour according to animals' needs.

## Author summary

Animals and humans can search for different resources depending on their needs. For example, when you are thirsty at work, you may go to a common room where hopefully coffee or water is available, while if you are hungry, you would rather go to a canteen. Such ability to seek different resources based on a physiological state is so fundamental to survival, that is present also in simple animals. This paper proposes how this ability could arise from a simple neural circuit that can be mapped on evolutionary older parts of the

available at https://github.com/YuhangSong/reward-bases.

**Funding:** This work has been supported by Biotechnology and Biological Sciences Research Council (https://www.ukri.org/councils/bbsrc/) grant BB/S006338/1 (B.M., Y.S., M.W. and R.B.), Medical Research Council (https://www.ukri.org/councils/mrc/) grant MC_UU_00003/1 (R.B.), Wellcome Senior Research Fellowship 202831/Z/16/Z (M.E.W.), Henry Dale Fellowship from the Wellcome Trust (https://wellcome.org/) (A.L.) and Royal Society (https://royalsociety.org/) 213465 (A.L.). The funders played no role in the study design, data collection and analysis, decision to publish, or preparation of the manuscript.

**Competing interests:** The authors have declared that no competing interests exist.

vertebrate brain, called the basal ganglia. The model suggests that this circuit learns the availability of different reward types, and then combines them according to the physiological state to control behaviour.

# 1 Introduction

Neuromodulator dopamine plays a key role in reinforcement learning (RL). The dominant theory of its function is the Temporal Difference (TD) learning model, according to which the dopamine neurons in the Ventral Tegmental Area (VTA) encode reward prediction errors which are used to modulate plasticity at cortico-striatal synapses so as to learn expected reward at different environmental states [1–3]. This theory is supported by large amounts of experimental data showing that dopamine neurons produce patterns of activity consistent with reward prediction errors [4–7] and modulate cortico-striatal plasticity in the directions predicted by the theory [8, 9]. However, a key assumption of this model is that the amount of reward from different outcomes is fixed, while for biological organisms the 'reward function' can fluctuate over time depending on physiological state—e.g. food is rewarding when hungry but not when satiated. Throughout this paper we distinguish between two aspects of animal's state: physiological state such as hunger or thirst, and environmental state such as animal's location in space or a stimulus presented during an experiment.

There is substantial evidence that animals can flexibly adapt their behaviour in the face of changes to physiological needs, and they can instantly adapt their preferences without having to experience the rewards in the new physiological state. A compelling demonstration of this capability comes from a series of experiments [10] involving rats which learn to associate two levers with receiving either pleasant sugar juice or extremely unpleasant salt water. When they were placed in a physiological state which mimics the brain chemistry of salt deprivation, which they have never experienced before, they immediately approached the lever which was associated with the salt water. This cannot be explained through standard RL algorithms such as TD learning which require an experience of reward to update the value function estimate. TD learning would predict that the salt-deprived animals would still avoid the salty lever until the salt solution is delivered which would give them an unexpectedly positive reward, and would slowly induce them to approach that lever more and more often until it becomes the primary lever to be approached. Beyond this classic demonstration, there is a wide range of literature arguing that Pavlovian learned associations appear to be dynamically responsive not just to experienced rewards but to internal physiological states [11–14], and it has been demonstrated that stimulation of particular neurons in hypothalamus reliably produces drinking behaviour within a few seconds, even in fully water-satiated animals [15].

There have been several explanations proposed for this instant reward revaluation capability in the literature. It has been proposed that this ability is due to model-based planning [16, 17]. Following Daw et al. [16], we use the term 'model-based RL' to refer to algorithms that "learned, over experience, the structure of action-induced [environmental] state transitions in the task" (p.1707) and use it to evaluate consequences of considered actions. Another approach is to use successor representations [18] which represent a 'successor matrix' of discounted environmental state occupancies allowing value functions to be estimated for any given reward function. However, these approaches are computationally complex and thought to be implemented in the cortex [19, 20], while controlling behaviour based on physiological state is so fundamental for survival that it is present in animals without cortex, such as drosophila where it is mediated by dopamine neurons [21]. In vertebrates dopamine neurons innervate the basal

ganglia system, which is a core component in reward-based action selection [3, 22–26], and is linked to brain systems monitoring physiological state such as the hypothalamus [27–30]. The ability of basal ganglia to control behaviour based on physiological state is consistent with observations that the dopamine release depends on physiological state [31, 32], as well as that dorsomedial striatal lesions abolish flexible reward devaluation behaviour [33]. Moreover, there is evidence that the basal ganglia model-free system may be relied on *more* than the model-based system when physiological needs are pressing [34]. There is thus a fundamental difficulty with the standard theory: the basal ganglia needs to flexibly adapt to changing reward functions to effectively control behaviour, yet TD learning, the algorithm it supposedly implements, simply cannot do this.

Recent data suggest that dopamine neurons may support computations beyond the standard TD learning. It has been observed that, in addition to reward prediction error, dopamine neurons encode information about diverse variables, such as movement [35, 36] or sensory features [37]. Moreover, it has been recently demonstrated that distinct mammalian dopamine neurons preferentially respond to food or water [38]. In insects, the selectivity of dopamine neurons for different reinforcement types has been analysed systematically, and it was observed that different dopamine neurons respond to different reinforcements such as sugar [39], water [40], courtship [41] and aversive stimuli [42].

In this paper, we propose a simple extension to TD learning that allows changes in physiological state to instantly modify the assigned values of environmental states, and hence behaviour. Specifically, we demonstrate that if we model the reward function as a linear combination of *reward basis vectors* and then learn a separate value function for each reward basis using standard TD learning, then we can instantly compute the value function of any reward function in the span of the reward basis vectors. Moreover, this algorithm can be implemented in neural circuitry in which dopamine neurons encode multiple prediction errors associated with different reward dimensions. Our model predicts that individual dopamine neurons should be selective for some reward dimensions more than for others, and we conduct a preliminary investigation of this prediction by reanalysing a small existing dataset recorded from dopamine neurons of a single primate [43]. Additionally, we demonstrate that our model can reproduce instant generalisation to new physiological state seen in dopamine responses [31] and behaviour [10], and achieves similar performance as successor representations [18] at RL tasks while using much less memory.

## 2 Results

### 2.1 Reward Bases model

Following standard RL, we denote the current environmental state of an agent by $x$. We also denote the reward received in the current environmental state by $r(x)$. As in standard TD model, we assume that the goal of the learning process it to estimate the value function of environmental states defined as:

$$\mathcal{V}(x) = E(r(x) + \gamma r(x') + \gamma^2 r(x'') + \cdots) \tag{1}$$

In the above equation, $E$ denotes an expectation, $x'$ denotes a environmental state in which the agent finds itself in the next time step, and $\gamma$ denotes discount rate expressing how much a reward in the next time step is worth to the agent relative to the same reward in the current time step. Thus the value function $\mathcal{V}(x)$ expresses how much reward is expected in environmental state $x$ immediately and into the future, while considering that reward in the future is worth less than the reward now.

The Reward Bases model extends standard RL by introducing three assumptions. First, based on models of homeostatic regulation [44–46], we assume that the physiological state the animal seeks to optimise has multiple dimensions. In particular, we assume that the reward can be decomposed into a linear combination of component rewards, or *reward bases*:

$$r(x) = \sum_i m_i r_i(x) \tag{2}$$

where $r_i(x)$ denotes an individual reward basis, and $m_i$ is a motivational drive parameter for each reward basis that determines how outcomes are mapped into their utility [47]. For instance, if $r_i(x)$ is a reward basis which represents food, then $m_i$ would reflect the degree of hunger, describing how much food is valued by the agent. We assume that the basal ganglia system represents those dimensions of reward that are most important for survival (the evidence shown later suggests they include food and water, but the exact range of dimensions will need to be identified by future studies).

Second, as in models of multi-objective RL [48, 49], we assume that animals learn separate value functions $\mathcal{V}_i(x)$ associated with individual reward bases. These value functions are learnt using an analogous rule as in the TD learning, but using the reward basis as the reward instead of the full reward,

$$\Delta \mathcal{V}_i(x) = \alpha \delta_i(x) \tag{3}$$

$$\delta_i(x) = r_i(x) + \gamma \mathcal{V}_i(x') - \mathcal{V}_i(x) \tag{4}$$

where $\alpha$ denotes the learning rate, and $\delta_i(x)$ is a prediction error associated with reward basis $i$, defined analogously as in standard TD learning (cf. Eqs 10 and 11 in Methods) but for a single reward dimension.

Third, the key novel assumption of our model is that the animal computes the total value function by combining the value functions $\mathcal{V}_i(x)$ weighted by their *current* motivational drives

$$\mathcal{V}(x) = \sum_i m_i \mathcal{V}_i(x) \tag{5}$$

Eqs 3 and 4 describe learning in the fundamental version of the Reward Bases model, and later in the paper it will be shown how these rules can be further refined to capture specific experimental data, but for simplicity we start with analysing this version.

We now describe two key properties of the Reward Bases model. The first property is that if the motivational drives $m_i$ are constant in time, the Reward Bases model computes the same value as the TD learning model. This can be shown by substituting the definition of reward bases (Eq 2) into that of the value function (Eq 1),

$$\mathcal{V}(x) = E\left( \sum_i m_i r_i(x) + \gamma \sum_i m_i r_i(x') + \gamma^2 \sum_i m_i r_i(x'') \cdots \right) \tag{6}$$

$$= \sum_i m_i E(r_i(x) + \gamma r_i(x') + \gamma^2 r_i(x'') \cdots) \tag{7}$$

$$= \sum_i m_i \mathcal{V}_i(x) \tag{8}$$

In the last transformation we used $\mathcal{V}_i(x) = E(r_i(x) + \gamma r_i(x') + \gamma^2 r_i(x'') \ldots)$ as $\mathcal{V}_i(x)$ are learned analogously to TD learning so converge to such values.

The second key property of the Reward Bases model seen in Eq 8 is that the value function decomposes into a weighted sum of component value functions, to which we refer as a *value bases* since they are the value functions of specific reward bases. Given a set of value bases, and a set of coefficients $\{m_i\}$, we can instantly compute any value function spanned by the reward bases. Hence, when the animal's physiological state changes, this enables instant generalisation of the optimal value function for *any* reward function expressible as a linear combination of the reward bases.

To build intuition for how the Reward Bases model functions, in Fig 1 we present a visualisation of the value functions learned by a TD and Reward Bases agents in a simple spatial navigation task. The agents must navigate around a $6 \times 6$ grid where they can obtain rewards in three locations (indicated in dark blue in the top row). We assume that these three rewards are of different types, and activate separate reward bases (Fig 1B top). The TD agent received a total reward defined as $r(x) = \sum_{i=1}^{3} r_i(x)$ (Fig 1A top). The value function learnt by the TD and Reward Bases agents are shown in the middle row of Fig 1. Using the set of value bases in Fig 1B and equal weighting coefficients of $m_i = 1$, the Reward Bases agent can exactly reconstruct the total value function of the TD agent (cf. Fig 1A middle and Fig 1B middle right). Importantly, the Reward Bases agent can also instantly generalise to other reward preferences. For instance, when only the first reward becomes fully valuable, the second only keeps half of

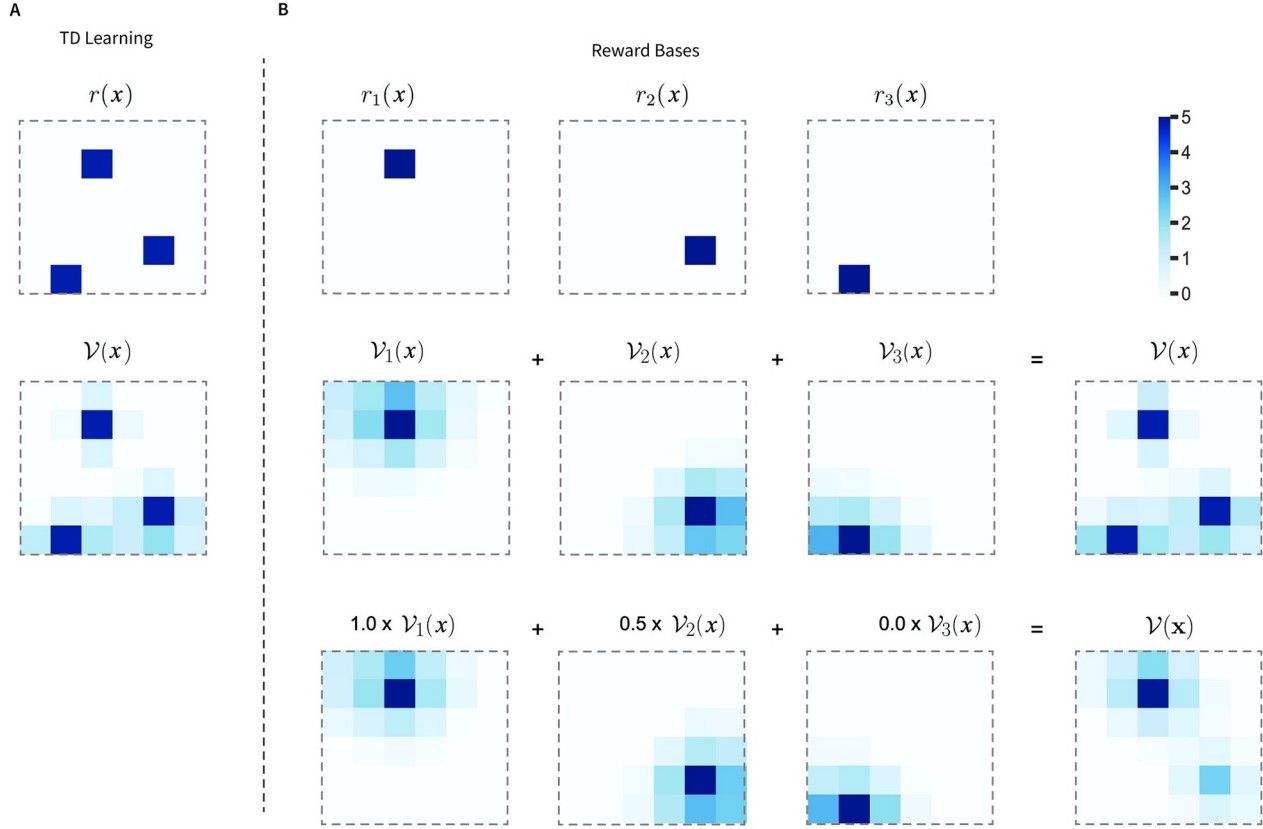

**Fig 1. Visualisation of the value functions learned by the models. A:** TD model. **B:** Reward Bases model. The task consisted of a $6 \times 6$ grid with three items. The Reward Bases agent used a separate reward basis for each object, giving it $+5$ for that object, and $-0.1$ for all other squares. The value functions displayed are obtained by the agents exploring the environment for 1000 steps with a random action policy, learning rate $\alpha = 0.01$ and discount factor $\gamma = 0.99$.

its value, and the third is not valuable, the Reward Bases agent can choose its actions according to the appropriate value function shown in Fig 1B bottom.

## 2.2 Neural implementations

Thus far, we have described the Reward Bases model at an algorithmic level, now we will consider different possible implementations of this algorithm in the circuitry of the basal ganglia. They can be achieved by modifications of the original circuit model [55] illustrated in Fig 2A. In this implementation of TD learning, dopamine neurons in the VTA receive two inputs—the incoming reward and the temporal derivative of the value function (that can be provided by inputs from the striatum [56]). To compute the reward prediction error, the dopamine neurons simply need to add these two inputs. These reward prediction errors can then be sent to the ventral striatum where they are used to update synaptic weights encoding the value function. This circuit can be extended to implement the Reward Bases model in two ways, which we present below, and later compare with experimental data.

A simple mapping of the Reward Bases algorithm on a neural circuit is shown in Fig 2B, and the only change is that instead of assuming a single homogeneous population of dopamine neurons responding to a global reward signal, we instead assume that neurons encoding values and prediction errors are parcellated into groups, each responding to a single reward basis. This reward basis parcel is simply the same circuit as original model in miniature in that dopamine neurons simply receive both the reward for that basis, and the temporal derivative of the value function for that basis and compute the reward prediction error for that basis, which is then sent specifically to the region which requires it. A set of parallel units represents the value

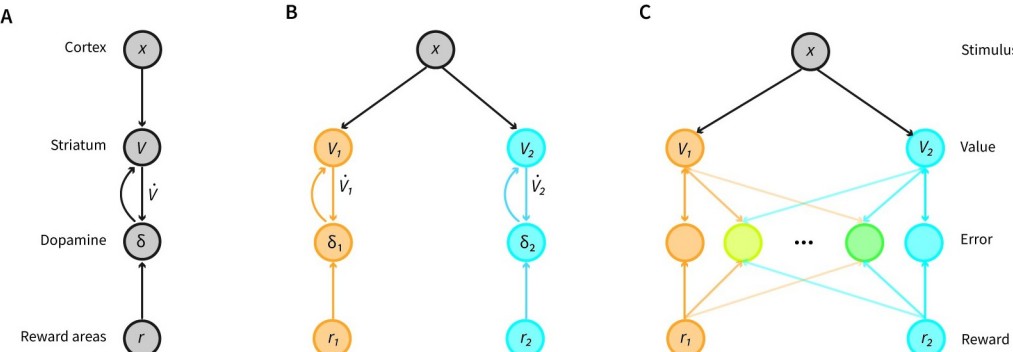

**Fig 2. Possible neural implementations of TD learning and the Reward Bases model. A**: Standard TD learning model. Dopamine neurons receive reward input and compute a reward prediction error. This reward prediction error is projected to the striatum where it modulates plasticity of the cortico-striatal synapses that learn a value function. In the diagram, the environmental state is assumed to come from the cortex, because in tasks where the environmental state is determined by the stimulus, on which we focus in this paper [10, 31, 43], the information about the stimulus is mainly carried by the sensory cortex. However, please note that this is a simplification as the cortex also encodes many other quantities including value [50], and in more general tasks the environmental state may need to be inferred and information about the environmental state may be brought by other regions (e.g., the hippocampus). The value is assumed to be computed in the striatum following the convention typically used in RL models [51, 52], but note that this assumption is a matter of debate. In particular, a study [53] questioned the validity of the previous analyses of value coding in the striatum as it demonstrated these analyses did not correctly account for temporal correlations in the data. Nevertheless, it was later demonstrated that after accounting for these correlations, the value signals can be found [54]. **B**: Single selectivity implementation. This requires parallel neural populations computing prediction errors and 'value function' neurons with the same connectivity patterns as the standard model. **C**: Mixed selectivity implementation. Reward Bases model can be approximated with dopamine neurons selective for a mixture of prediction errors. The saturation of colour of orange and blue lines corresponds to the strengths of the connections, and the gradation of colour of dopamine neurons indicates their degree of selectivity for the two reward bases.

bases, from which the ultimate value function can be computed through a simple linear combination with the weighting coefficients. We refer to this architecture as "single selectivity implementation", because in this circuit, the dopamine neurons are selective for a single dimension of reward. This neural implementation is related to previous models that also assumed multiple parcels of striatal neurons connected with corresponding dopamine neurons [57, 58], but the key difference is that in the single selectivity implementation the different modules are weighted according to the animal's physiological state and this fact is exploited to enable instant generalisation of the value function and ultimately behaviour to different physiological states.

The Reward Bases model can also be approximated in a circuit shown in Fig 2C, where the dopamine neurons are selective for different mixtures of prediction errors. We refer to it as "mixed selectivity implementation", and the details of the model will be introduced later. We later demonstrate that such distributed coding of prediction errors by dopamine neurons also enables learning of the correct value bases.

Both implementations of the Reward Bases model predict that the activity of individual dopamine neurons may not only depend on the overall value but also on the type of reward. Additionally, the single selectivity implementation predicts that individual dopamine neurons should signal reward prediction errors for a specific reward basis, while the mixed selectivity implementation predicts that while some neurons may encode reward prediction error for a single dimension, others may encode a combination of prediction errors. Below we conduct a preliminary comparison of these predictions with experimental data.

## 2.3 Responses of dopamine neurons to stimuli predicting different reward types

Testing the above predictions requires a unique experiment in which individual dopamine neurons are recorded in a task where stimuli are associated with delivery of different types of reward. To our knowledge there exists only one published study that obtained such data—by Lak et al. [43]. This study used different juices and reward risks in two monkeys to demonstrate encoding of subjective economic value by dopamine neurons (77 neurons analysed), and verified the results in a juice vs food experiment in only one monkey (from which 19 neurons were analysed). The juice vs food experiment better examines and separates representation of subjective economic value and different reward dimensions, and is therefore suitable for testing model's predictions. Keeping in mind that the conclusions drawn from 19 neurons from a single monkey require further verification, we nevertheless feel it is worth presenting the analyses of these unique published data to provide a preliminary test of the theory, and inspire future studies.

In the experiment, a monkey was presented with 5 stimuli associated with different volumes of juice or different amounts of banana (Fig 3A). In the first part of the experiment, on each trial, the monkey was making a choice between two stimuli, and obtained corresponding rewards (which could be juice or banana). Choices on these trials were used to estimate the subjective utility of rewards associated with each stimulus. In the second part of the experiment, on each trial the monkey was presented with a single stimulus, and then, after 1.5s delay, received the corresponding reward. During these trials, the activity of dopamine neurons was recorded. Fig 3B displays the subjective utility of the 5 rewards estimated by Lak et al. [43] from the first part of the experiment. The monkey exhibited a clear preference between different amounts of juice and banana rewards. The utility increased with the amount of reward, and the two most preferred rewards were the largest volume of juice and the largest quantity of banana.

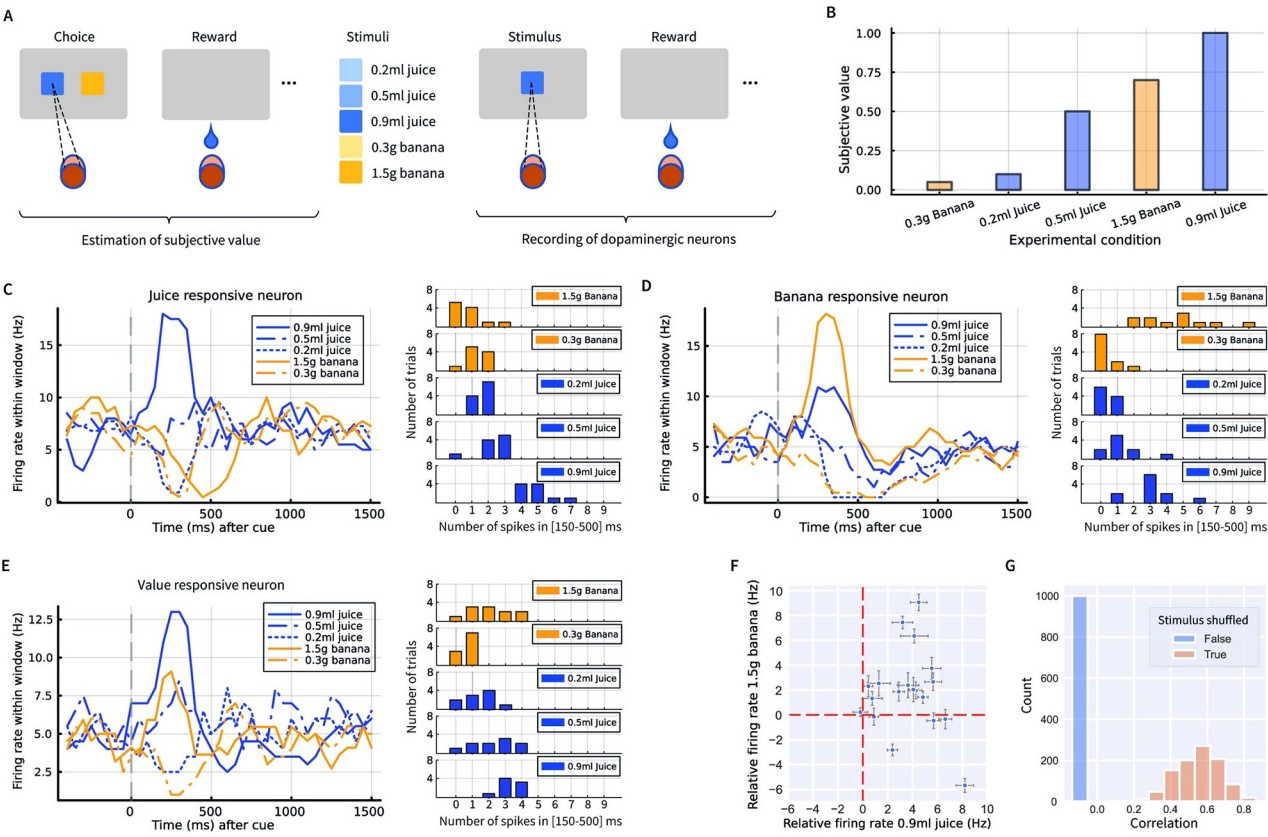

**Fig 3. Analysis of responses of individual dopamine neurons. A**: The experimental paradigm [43]. During recording, a monkey was presented with a visual stimulus indicating the reward type. We analysed the responses of individual dopamine neurons in the period after the stimulus presentation. **B**: The subjective value determined by Lak et al. [43] for each of the 5 conditions which could be presented to the monkey. **C, D, E**: Average neural firing rates during a trial for a juice responsive neuron, a banana responsive neuron, and a value responsive neuron. The left displays show the firing rate as a function of time within a trial (see Methods Section 4.2), where time 0 corresponds to onset of the stimulus indicating which reward type will be presented. To visualise the variability of the responses to different stimuli across trials, the right displays show histograms of the number of spikes within a window of 150–500ms after stimulus onset for each condition for the corresponding neuron. **F**: Responses of individual neurons to stimuli predicting the largest food or drink rewards. Each dot corresponds to a single neuron, and its coordinates correspond to the average responses to either 0.9ml of juice (x-axis) or 1.5g of banana (y-axis). The response is computed as the firing rate within a window 150–500ms after stimulus onset relative to the baseline 0–500ms prior to the stimulus onset. The error bars show the standard error of the mean. **G**: Correlation between responses to 0.9ml of juice and 1.5g of banana in experimental data (blue), and when repeating the analysis 1000 times with shuffled trials (orange).

We re-analysed the activity of individual dopamine neurons, and found that, indeed, a fraction of recorded neurons are differentially selective towards the prediction errors associated with juice or banana. Fig 3C shows an example neuron whose activity is significantly modulated by the volume of juice predicted by the stimulus but little by the amount of banana. In particular, the firing rates differ substantially between stimuli predicting different volumes of juice, while they are similar for large and small amounts of banana. Remarkably, the firing rate of this neuron decreases below baseline after cue predicting 1.5g of banana even though it was the second most favourite reward (Fig 3B). Such a decrease is predicted by single selectivity implementation for a dopamine neuron encoding prediction error associated with liquid, because the banana provides less liquid than the average across the rewards in the experiment. Nevertheless, the existence of such neurons is also admitted by the mixed selectivity implementation.

Fig 3D shows the activity of a neuron which is modulated by the amount of food predicted by the stimulus more than by volume of juice. For this neuron, the responses to stimuli predicting large and small amounts of food are separated more than responses to different volumes of juice. Nevertheless, this neuron is modulated by both reward types to a certain extent, so it does not conform to the prediction of the single selectivity implementation, which assumes prediction errors for different reward types to be encoded by separate dopamine neurons, hence these data are more consistent with the mixed selectivity implementation.

Fig 3E shows an example of a neuron whose activity is modulated just by the subjective value and does not depend on reward type, i.e. the magnitude of responses to different stimuli in Fig 3E follows an order determined by their subjective value in Fig 3B. Such coding of overall value by dopamine neurons has been pointed out in the original study presenting these data [43]. Interestingly, this pure value neuron was recorded on the same day as the juice-modulated neuron in Fig 3C, which suggests that different sensitivity to reward types cannot be explained by fluctuation of animal's preferences for different reward types across days. Activity of all recorded neurons including the three neurons shown in Fig 3C, 3D and 3E is visualised in S1 Fig.

To visually illustrate the diversity of responses of dopamine neurons, Fig 3F shows the responses to the stimulus predicting the largest amount of banana, against the stimulus predicting the largest volume of juice. If dopamine neurons encoded single reward prediction error signal from the standard TD model, then their responses to these two stimuli would be similar and positive, because these two rewards are preferred by the animal more than other outcomes, hence would produce positive prediction errors. By contrast, dopamine neurons produced diverse responses, and two neurons even substantially decreased their firing rates after the stimulus predicting 1.5g of banana. Furthermore, if dopamine neurons encoded unified reward prediction error as expected in the conventional theory, the responses to these stimuli would be positively correlated. By contrast, the correlation is close to 0 and even slightly negative, as shown in Fig 3G.

We then compared this result with the correlation expected from a null hypothesis that the responses of dopamine neurons are the same after the stimuli predicting the largest amount of banana and juice. To obtain a null distribution, we recalculated the correlation 1000 times: In each iteration we randomly separated trials with stimuli predicting the largest amount of banana and juice into two groups, calculated the averaged response magnitude of each trial group for each neuron, and then calculated a correlation between responses on the two trial groups across neurons. The distribution of such correlations is shown in Fig 3G, and in all 1000 repetitions the correlation obtained from the shuffled data was larger than from the experiment, suggesting that the experimental correlation is significantly lower than expected from the above null hypothesis ($p<0.001$).

To provide another illustration for the diversity in selectivity of dopamine neurons, for each dopamine neuron we fitted a regression model that predicted firing rate after the stimulus onset based on prediction error for the two reward bases. The obtained regression coefficient are plotted in Fig 4A. If dopamine neurons encoded a single reward prediction error signal as predicted by the standard TD model, then the regression coefficients for the two prediction errors should be equal, and the points in the plot should be concentrated along the diagonal identity line. By contrast, several points are distant from the diagonal. Fig 4A also demonstrates it is possible to distinguish between the predictions of single selectivity and mixed selectivity implementations. The single selectivity implementation predicts that each of dopamine neurons encodes single dimension, thus should have the regression coefficient for the other dimension close to 0, and so the points on this graph should be accumulated along either the vertical or horizontal axis. Although there are some

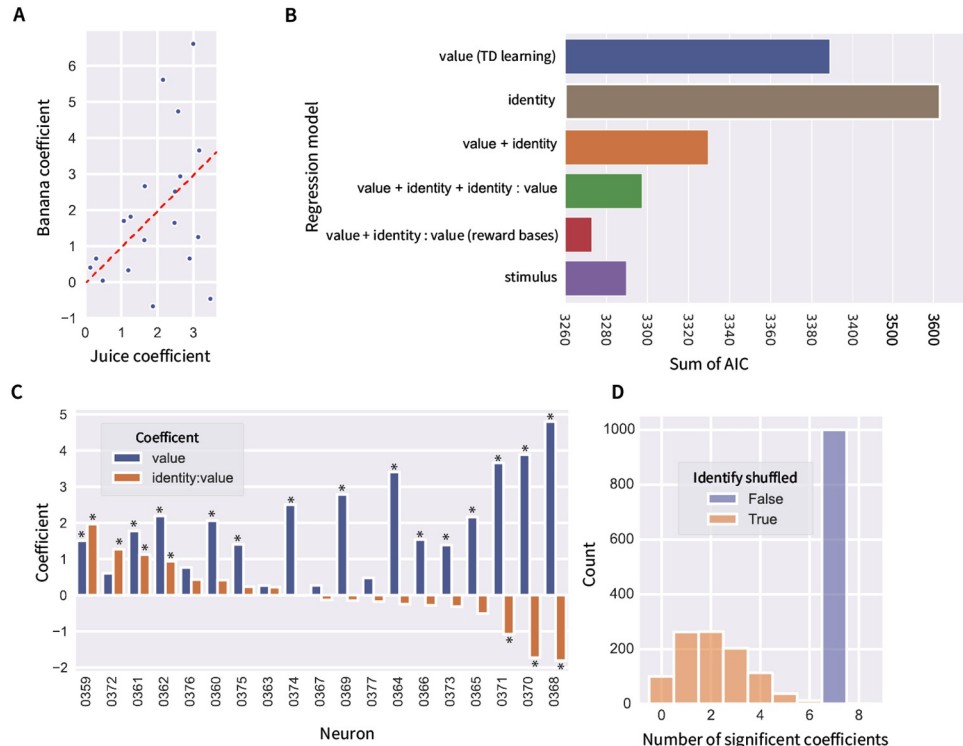

**Fig 4. Regression analysis of activity of dopamine neurons. A**: Regression coefficients indicating to what extend individual dopamine neurons encode prediction errors associated with the two reward bases. Each dot corresponds to one neuron. Its coordinates are the regression coefficients estimated from biological neurons. The dashed red line shows the identity line on which the points should lie according to the TD-learning model. **B**: Comparison of the Akaike Information Criterion (AIC) of regression models predicting the activity of dopamine neurons. **C**: The coefficients for a linear regression predicting the number of spikes in the post-stimulus period as a function of the subjective value and the interaction of subjective value and reward type. Stars indicate significant coefficients ($p < 0.05$ uncorrected, t-test for significance of regression coefficient, two-sided). **D**: Comparison of the number of neurons with significant coefficient of interaction between identity and value in the data and when repeating the analysis 1000 times with shuffled assignments of reward identity across trials.

neurons with coefficients for one dimension close to 0, overall the points are spread out suggesting that the majority of dopamine neurons encode different mixtures of prediction errors.

To quantify the extent to which the activity of each dopamine neuron is dependent on subjective value and identity of reward, for each neuron, we fitted a set of regression models that predicted each neuron's firing rate after stimulus onset based on subjective value and identity regressor (equal to 1 for juice and to −1 for banana). The models are listed in Fig 4B and include a basic regression just using value, as well as models using both value and identity, as well as their interactions. The model from Fig 4A assuming that dopamine neurons encode a combination of prediction errors associated with different reward bases is equivalent to the model employing subjective value and its interaction with identity, as explained in the Methods Section 4.2. This model provided the most parsimonious description of dopamine firing (lowest Akaike Information Criterion) (the same conclusion was obtained using Bayesian Information Criterion, S2 Fig). The interaction of identity and value is evident in Fig 3D showing that this dopamine neuron was not generally more active for banana, but rather its activity was more modulated by value for stimuli predicting banana than juice.

To exclude a possibility that the diverse responses of dopamine neurons are driven by neurons preferring particular visual stimuli (presented in the experiment) rather than type of reward, we also compared a regression model employing the stimulus presented. This regression model contained 5 parameters (corresponding to the mean firing rate of a neuron to the 5 stimuli). Fig 4B shows that this model provides a less parsimonious description of dopamine activity than the model employing value and its interaction with identity. To verify if the Akaike Information Criterion can distinguish between these two models based on the amount of data we had available, we performed a model recovery analysis (S3 Fig). It shows that the observed difference between the Akaike Information Criterion for these two models is very unlikely for the surrogate data generated based on the model employing the stimulus. This suggests that the particular shape of visual stimulus was less responsible for driving dopamine activity than the value and type of reward predicted by the stimulus.

Fig 4C shows the estimated regression coefficients of the best fitting model for all neurons. Blue bars show the coefficients for value, which are all positive, implying that the activity of all neurons is affected by the value. The orange bars show the coefficients of interaction between identity and value. A positive coefficient indicates that the corresponding neuron is more modulated by value for juice, while a negative coefficient indicates larger modulation for banana. We found that 7 neurons out of 19 possessed a statistically significant interaction coefficient implying that their response depended on reward type. To investigate if this number of significant interaction coefficients could arise by chance, we compared it with the number expected from a null hypothesis that the responses of dopamine neurons do not depend on the type of reward predicted by the stimulus. To obtain a null distribution we repeated the following analysis 1000 times: In each iterations we shuffled assignments of trials to the reward type (i.e., we replaced the identity regressor across all trials recorded for a given neuron by its random permutation), performed regression predicting dopamine response from value and its interaction with (shuffled) identity, and counted the number of neurons with significant interaction coefficient. Fig 4D shows that in all 1000 repetitions the observed number of neurons with significant interaction coefficient was smaller than in the experimental data, so the probability of obtaining 7 significant neurons under the null hypothesis is $p < 0.001$. In sum, our analysis suggests that the activity of dopamine neurons is modulated by subjective value, as reported in the original study [43], and additionally, there is also a gradient for coding reward type across population, so that some dopamine neurons have responses modulated by different reward dimensions.

We have also investigated if the selectivity of dopamine neurons for reward type was stable during the recording session. S4 Fig shows that there is a strong correlation between the identity-value interaction coefficients estimated from the first half of trials and the second half, suggesting that the dopamine neurons had similar preferences for reward type throughout the recording session.

In summary, our analysis of the experimental data provides a preliminary support for the prediction of the implementations of Reward Bases model, that different dopamine neurons are selectively modulated by different reward dimensions. The data does not support the single selectivity implementation, where each neuron should encode a single reward basis, but rather is consistent with the mixed selectivity implementation, because the neurons appear to be more or less selective for a reward type, but not entirely unselective for the other. Later in the paper, we will present a network model which implements mixed selectivity, and demonstrate that the mixed selectivity of dopamine neurons is sufficient for learning value bases, but before this, we introduce an assumptions on dependence of dopamine activity on physiological state that will be used in the network model, and other simulations in this paper.

## 2.4 State-modulated prediction errors

In the model described at the start of the Results section, the reward prediction errors are still computed and the value basis is still updated even if that basis is not valued at the current moment. This is mathematically optimal given no computational or resource constraints since it enables flexible and instant generalisations to the greatest possible range of reward revaluations. However, there is a growing body of research that suggests that this is not the case in the brain since animals only appear to learn value functions of tasks when they are important to the animal and thus have motivational salience. For instance it has been shown that when fish were first exposed to a stimulus when hungry, they chose it substantially more often over an equally palatable alternative than when they were exposed to the stimulus when sated [59]. This implies that there is a direct modulation of the values learnt during training based on the current physiological state which is maintained even after the physiological state has changed. Similarly, it has been shown that physiological state can modulate dopaminergic prediction error responses [32]. In that study if the animals were trained to receive food rewards after a cue in a 'depleted state' (i.e. they were hungry), then during testing dopamine neurons respond to the cue (CS) and not the food reward (US) as in the classic results [3]. However, if the animals were trained in a sated state, then when tested, the dopamine neurons responded only to the food reward (US), 'as if' they had not learned the task at all.

The version of Reward Bases model described at the start of Results cannot explain these findings since it decouples dopaminergic activity—and hence value function learning—from the physiological state such that the physiological state is only used to dynamically modulate the weighting of the value function bases during action evaluation. However, it has been shown that such effects can be straightforwardly explained by redefining the dopaminergic prediction errors such that they are also modulated by the physiological state [60]. In particular, a state-dependent prediction error associated with reward basis $i$ is defined as

$$\tilde{\delta}_i \quad = m_i \delta_i \tag{9}$$

In the Methods Section 4.3 we describe such a model with state-dependent prediction error, in which $\tilde{\delta}_i$ also drives synaptic plasticity. Effectively, this means that the 'learning rate' of the TD update is modulated by the physiological state variable such that animals learn much more rapidly when they are 'depleted' and much less rapidly when they are sated. It has been demonstrated that such state-modulated prediction error can also be viewed as error in prediction of subjective utility of rewards given specific mathematical assumptions about the form of the utility function [60]. Note that this modification does not affect predictions tested or simulations earlier in the paper, because in the experiment of Lak et al. [43] the animals were motivated to acquire both food and fluid (so $\tilde{\delta}_i = \delta_i$ for $m_i = 1$), and in the simulation in Fig 1 the learning took place while $m_i = 1$.

## 2.5 Learning with mixed selectivity

This section proposes a possible implementation of the Reward Bases model in a striato-dopaminergic circuit in which dopamine neurons encode mixtures of prediction errors in different dimensions. In this section, we make two contributions. First, we propose a hypothesis on how the striato-dopaminergic circuit could weight value bases by motivational drives to compute the value function. Second, we demonstrate that it is feasible to learn value bases even if dopamine neurons encode mixtures of prediction errors and send projections with random connectivity to the striatum. This second contribution is based on work [61] showing how a model of motor parts of striato-dopaminergic circuit can learn multi-dimensional action

signals. We demonstrate that a similar network can learn the correct value bases in the task of Lak et al. [43]. We summarise the model in this section, and provide a detailed description in Methods Section 4.4.

We consider a mixed selectivity model with the architecture shown in Fig 5A. To simulate dopamine neurons encoding different mixtures of state-dependent prediction errors, different dopamine neurons received inputs from neurons encoding reward of distinct types $r_i$ scaled by the corresponding motivational drives $m_i$, through connections with randomly generated strengths $w^{R \to D}$. We did not wish to assume any structured connectivity between dopamine and striatal neurons, hence the strengths $w^{D \to S}$ of these connections was also randomly generated. This resulted in striatal neurons learning to encode different mixtures of value bases.

In order for the total output from all striatal neurons to encode the value function, the activity of striatal neurons encoding particular value bases need to be scaled by the corresponding motivational drives, hence a question arises of how the information on motivational drives could be delivered to appropriate striatal neurons. A possible answer to this question is hinted by a property that a subset of striatal neurons encoding a particular value basis is defined by dopaminergic input, i.e. striatal neurons that contribute to encoding a particular value basis are those that receive input from dopamine neurons encoding the corresponding prediction errors. Therefore, dopamine neurons are ideally positioned to also provide the weighting by motivational drive to the striatal neurons they project to. They can provide the information on motivational drives $m_i$, because their activity $\tilde{\delta}^k$ is scaled by $m_i$ (Eq 9). Consequently, we assume that the striatal neurons can decode motivational drives from the activity of dopamine neurons, and we denote by $D^k$ the mixture of motivational drives associated with prediction errors encoded by dopamine neuron $k$. One possibility for dopamine neurons to provide information on motivational drives is to encode $D^k$ in their tonic (slowly changing) activity, while encode prediction errors $\tilde{\delta}^k$ in phasic (burst) activity, as proposed in previous models

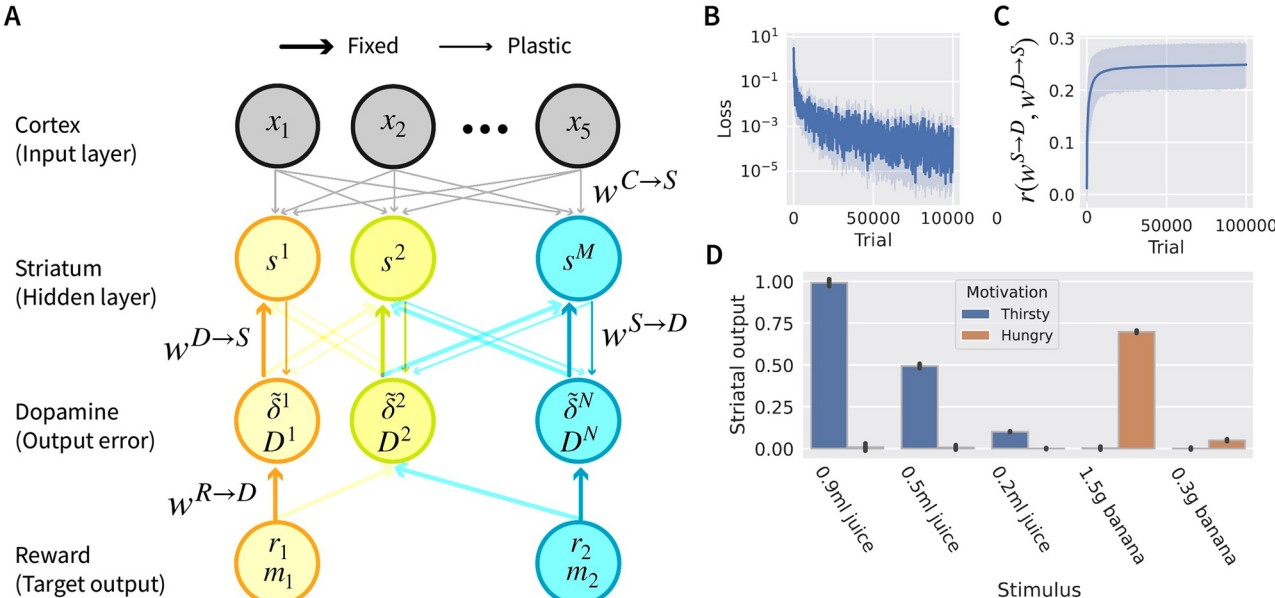

**Fig 5. Mixed selectivity model. A**: Architecture of the simulated network. **B**: Loss during training. **C**: Correlation between weights of striato-dopaminergic connections $w^{S \to D}$ and reciprocal projections from dopamine to striatal neurons $w^{D \to S}$. **D**: Total output from striatal neurons for different stimuli and motivational drives. In panels B-D, the error bars show standard deviation over $n = 10$ repeats of the simulation.

[51, 62]. We assume that the motivational drives encoded by dopamine neurons $D^k$ scale the gain of striatal neurons they project to (as in previous models [51, 62]), which enables value bases to be scaled by appropriate motivational drives, so that the total output from the striatum encodes the value function.

A key condition necessary for encoding of the value function in the total output from the striatum is the absence of prediction errors carried by dopamine neurons, because the lack of prediction errors $\tilde{\delta}^k = 0$ implies that the input on obtained rewards is matched by the striatal inhibition encoding predictions of expected reward (shown analytically in Methods Section 4.4). Therefore, synaptic plasticity in the model aims at minimising the prediction errors. Consequently, we defined the loss function (to be minimised through learning) as the sum of squared prediction errors of dopamine neurons.

The value bases in the model are encoded in a distributed fashion in the synaptic weights from the cortical to the striatal neurons $w^{C \to S}$ and from the striatal to the dopamine neurons $w^{S \to D}$, hence the network in Fig 5A could be thought of as a two layer neural network that transforms cortical input encoding an environmental state to the value of this state. Comparing this circuit to neural networks used in machine learning, the cortex corresponds to an input layer, the striatum to a hidden layer, the reward corresponds to the target output the striatum should predict, and dopamine neurons compute the error in the output of the network. The change in weights from striatal to dopamine neurons $w^{S \to D}$ (that could be considered as the "output-layer weights" in this network) can be found by gradient descent on the loss, which corresponds to Hebbian plasticity (shown in Methods Section 4.4). Changing the cortico-striatal weights $w^{C \to S}$ (that could be thought as "hidden-layer weights") according to the gradient of loss would correspond to the backpropagation algorithm [63], so would require neurons encoding the error in the output layer, i.e. the dopamine neurons, to send the error back to modulate the plasticity of hidden-layer weights. In agreement with this requirement, it is known that dopamine neurons modulate the plasticity of cortico-striatal synapses [8]. However, the backpropagation algorithm would additionally require the projections from dopamine to striatal neurons $w^{D \to S}$ to be equal to the reciprocal projections from striatal to dopamine neurons $w^{S \to D}$ (see Methods Section 4.4), while we did not wish to make such strong assumption and initialised these connections randomly. Nevertheless, it has been demonstrated [61] that the loss can be minimised in a similar network if the weights from dopamine to striatal neurons $w^{D \to S}$ are random and fixed, due to the phenomenon known as feedback alignment [64]. Therefore, in our model the plasticity of cortico-striatal weights $w^{C \to S}$ is modulated by dopamine activity received by the striatal neurons through fixed weights $w^{D \to S}$.

We simulated the network in a task corresponding to that studied by Lak et al. [43]. On each trial one of 5 stimuli was randomly chosen, motivational drives were set to random values $m_i \in [0, 1]$, and the corresponding reward was delivered. Fig 5B shows that the loss decreased during training. Fig 5C displays the correlation between striato-dopaminergic weights $w^{S \to D}$ and reciprocal weights $w^{D \to S}$ over training iterations. Recall that the correlation of 1 would be required for the training of cortico-striatal weights $w^{C \to S}$ with backpropagation algorithms. Fig 5C illustrates that during learning, the striato-dopaminergic weights $w^{S \to D}$ align themselves to reciprocal weights $w^{D \to S}$, so striatal neurons start to project more to the dopamine neurons they receive input from. Nevertheless, the correlation does not reach 1, possibly because a weaker alignment is sufficient for reducing the loss function. Fig 5D shows the average output from the striatum in trained networks. The blue bars show striatal output for different stimuli when the juice is valued ($m_1 = 1, m_2 = 0$), while the orange bars show the striatal output when the banana is valued ($m_1 = 0, m_2 = 1$). In each case, the striatal output is very

close to the subjective value of the presented stimulus (Fig 3B) if the reward type associated with the stimulus is valued (and to 0 otherwise). This demonstrates that the mixed selectivity model is able to learn the correct value bases for this task. Since we do not compare the model to data from individual neurons in the remainder of the paper, we will not include mixed selectivity in the remaining simulations for simplicity.

## 2.6 Dopaminergic activity reflects instant revaluation

Selectivity of individual neurons for different reward types analysed earlier in the paper was connected with one of the assumptions of the Rewards Bases model (learning value bases with separate prediction errors; Eq 4), but to account for these data it is not necessary to assume that the value bases are weighted according to physiological state (Eq 5). In this and the next sections we demonstrate that this additional assumption enables the Reward Bases model to capture instant revaluation after changes in physiological state seen in dopaminergic activity and behaviour, which cannot be described by the standard TD learning. Here, we focus on qualitatively matching the dopaminergic responses in a reward devaluation task in which rats could press levers associated with either a food or a sucrose reward under varying conditions of selective satiation [31]. The experimental data we modelled were taken from forced trials in which only one of the levers delivered rewards. A schematic of the experimental paradigm [31] can be seen in Fig 6A. At the onset of each forced trial, animals were presented with a cue indicating which of the two levers can be pressed to obtain a reward. Each lever typically delivered a particular type of reward, but on some trials it could give either four times the amount of reward (MORE condition), or the other reward type (SWITCH) condition. In the devaluation condition, the animals were fed to satiation in one of the reward types but deprived of the other. Fast-scan cyclic voltammetry was used to measure dopamine levels in the nucleus accumbens core, part of the ventral striatum.

To capture the experimental data [31], we employed a model with state-modulated prediction error, because such modulation is necessary to explain the experimental data. For example, Fig 6B shows responses to different rewards in MORE conditions in different physiological states in the task illustrated in Fig 6A. The dopamine level increases after receiving more food (or drink) than expected, only if the animal was hungry (or thirsty), suggesting that the dopamine neurons encode prediction errors scaled by the corresponding physiological need.

We simulated an agent interacting with the paradigm illustrated in Fig 6A. The agent was trained with two reward bases, one which gave out 1 reward for each food reward and 0 for sucrose, and a sucrose basis with the opposite reward schedule. To simulate testing in devaluation sessions where the animals were fed to satiety in one reward type but deprived of the other, the reward basis weights $m_i$ of the agent were fitted to the data to determine how the reward weights changed during devaluation. Since voltammetry measures relative changes in extrasynaptic dopamine, we use our model to make predictions of total dopamine release, which we take to be the sum of the dopamine neurons, i.e, $\sum_i \tilde{\delta}_i$, summing across the contributions of all reward bases.

A key experimental result [31] is shown by blue bars in Fig 6C, visualising the dopamine response to the lever extension on the *first trial* after devaluation, before the animals received any of the devalued reward following a lever extension. The response to a lever associated with the devalued reward is lower than for the valued option even on the first trial [31]. This finding cannot be accounted for by standard TD learning, because the prediction errors following a cue reflect the values of the cues, which in TD learning model are only updated following reinforcements, but since devaluation the animals did not receive any rewards. The Reward Bases

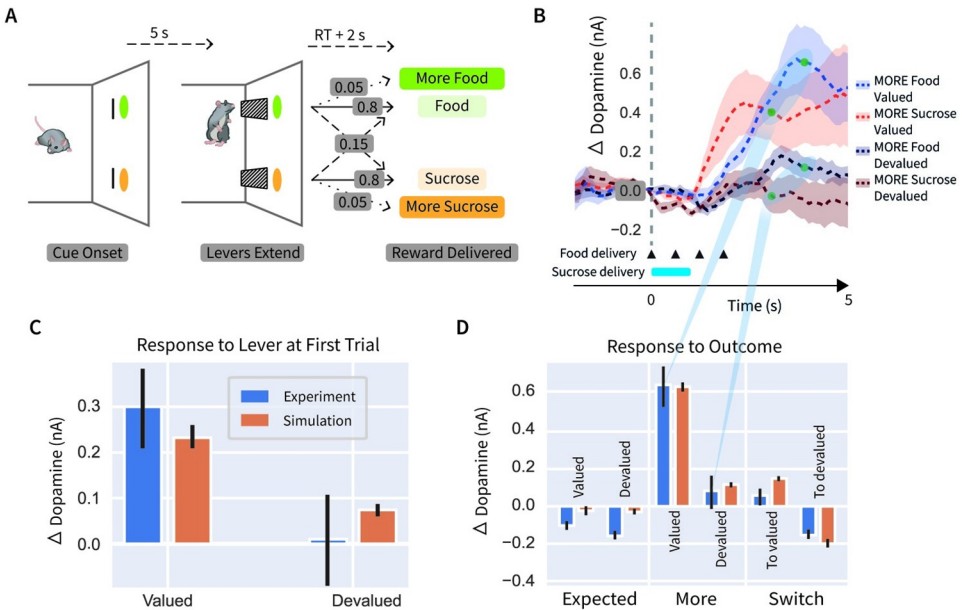

**Fig 6. Dependence of dopaminergic responses on physiological state. A**: Sequence of events during a trial in the experimental paradigm [31]. At the onset of forced trials, animals were presented with a single cue indicating which of the levers trigger reward delivery, while at the onset of choice trials both cues were shown. After a delay of 5s a single lever extends in the forced trials, or two levers extend in the choice trials (as shown in the figure). Following Reaction Time (RT, i.e. time to press a lever) and further 2s a reward is delivered. A particular type and size of the reward was delivered with a probability indicated by labels next to arrows. **B**: Dopaminergic activity in the MORE condition (re-plotted from Figure 2c in the original paper [31]). In this condition the reward delivery was extended in time: 4 pieces of food were delivered every 600ms as indicated by triangles, while sucrose was delivered for 1s as indicated by a blue bar. Model simulations were compared with dopamine levels 2s after the end of reward delivery (indicated by green dots), which were averaged across food and sucrose to give "experimental data" in panel D, as illustrated by the blue shading to panel D. **C**: Comparison of simulated dopamine level against data after extension of lever indicating which reward are likely to be available, taken on the *first trial* just prior to reward delivery. The experimental data corresponds to dopamine level 2s after lever extension (both mean and error bars showing standard error were read out from dashed curves in Figure 4a in the original study [31]). The error bars for the Simulation show standard error across 3 repetitions of the simulation (such small number of repetitions was sufficient as it already resulted in smaller error bars in simulation than in data). **D**: Comparison of simulated results against dopaminergic responses to reward delivery. The experimental data in EXPECTED and SWITCH conditions corresponds to dopamine level 2s after reward delivery (read out from Figures 2d-e in the original paper [31]). The error bars for the Experiment show the standard error aggregated over food and sucrose conditions, computed as $\sqrt{SE_{\text{food}}^2 + SE_{\text{sucrose}}^2}/2$ (where $SE_{\text{food}}^2$ and $SE_{\text{sucrose}}^2$ were read out from the corresponding figures in the original paper [31]).

model, however, straightforwardly predicts these results. This is because the weighting coefficient $m_i$ is lower for the devalued reward, thus the prediction error $\tilde{\delta}_i$ associated with the devalued option is also reduced, even before the devalued outcome is delivered.

Fig 6D shows that the Reward Bases model can capture key qualitative patterns seen in dopaminergic responses to the outcome. When the most common reward type was delivered, the observed dopaminergic responses slightly decreased, and in the model the sum of prediction errors was slightly negative, because the MORE or SWITCH trial did not occur. In the MORE condition, the dopamine response is much higher when the large amount of valued reward is delivered, and the model reproduces this pattern because the positive prediction error $\tilde{\delta}_i$ is scaled by a larger weight for the valued reward. In the SWITCH condition, the dopaminergic response is higher when valued rather than devalued reward is given, and the model reproduces this pattern because the positive prediction error caused by switch to valued dimension is scaled by a larger weight.

## 2.7 Behaviour reflects instant revaluation in novel physiological state

In this section we show that the Reward Bases model can account for instantaneous changes in behaviour following a novel physiological state, which cannot be described by the TD learning. As mentioned in the Introduction, a classic experiment demonstrated the ability to instantly revalue salt when it is depleted [10]. It utilised a Pavlovian association paradigm in which rats were repeatedly exposed to one of two cues (extension of one of two levers)—associated with either intra-oral delivery of sucrose (pleasing) or salt (aversive) solution. When rats were injected with aldosterone and furosemide which mimics severe salt deprivation, immediately responded positively to the salt cues (Fig 7A). These results imply that the rats clearly possess the ability to perform instant generalisation to update associations upon physiological change with no direct experience of the positive rewards associated with the change. This phenomenon cannot be explained by the TD model, since the animals never experience the salt solution as rewarding, and have no opportunity to update their value function. These results can, however, be directly explained by the Reward Bases model.

To simulate the Reward Bases model in this task, we make two assumptions. First, we assume the animals can sense the amount of salt $r_{salt}(x)$ received in environmental state $x$. Second, since salt is a physiologically important quantity [65], we assume the animals would maintain a 'hardwired' value basis $\mathcal{V}_{salt}(x)$ estimating the total (current and discounted future) amount of salt received in environmental state $x$, that animals would learn during the Pavlovian association phase. In simulations we include two environmental states corresponding to cues predicting sucrose and salt. It is important to emphasise that the model does not explicitly learn that the salt cue will be directly followed by a environmental state in which salt is delivered (as in model-based explanations), but instead it just learns the total amount of salt expected in the future following that cue.

Upon the injection of the aldosterone and furosemide, the Reward Bases model can dynamically modulate its salt value basis with its physiological state to perform instant revaluation of its associations with the salt lever. This can occur even in the absence of any positive reward signal obtained by experiencing the salt, since the weighting coefficients $m_i$ act directly on the value bases, and thus the value bases themselves do not have to be updated. We demonstrate this instant generalisation capability of the Reward Bases model by replicating the behavioural results obtained in an experiment [10].

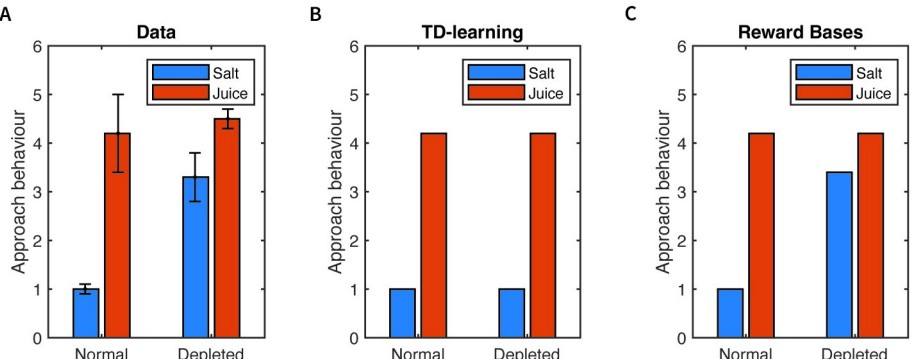

**Fig 7. Effects of salt deprivation on behaviour. A**: Experimental data showing approaches or appetitive actions towards the salt and juice lever in the homeostatic (normal) and salt-deprived conditions (data is replotted from Figure 3C in the study [10]). **B**: Simulation of TD learning. **C**: Simulation of Reward Bases model. In panels B and C, 'Approach behaviour' is equal to the estimated value of environmental states scaled by a constant chosen to match these values to the experimental data, as described Methods Section 4.6.

In simulation, the agents were exposed to interactions with the salt or juice levers (at random) and learnt a value function of being near the salt or juice levers. We assumed a linear proportional relationship between the learnt value function and degree of appetitive behaviour towards the lever. Fig 7B and 7C show that in the homeostatic (normal) condition, both TD learning and Reward Bases model are able to reproduce the behaviour of the animal, while in the sodium depletion condition, the Reward Bases agent can instantly generalise to match the behaviour of the agent, thanks to the weighting coefficients $m_i$ while the TD agent has no mechanism to vary its value function according to its physiological state, and so simply predicts that the salt lever will remain aversive to the animal in the sodium depleted condition. Although this simulated task is very simple, we choose to present it, because we feel that due its low complexity it best illustrates the difference between the models, and it is striking that the TD learning cannot account for behaviour in so simple task.

## 2.8 Relationship to successor representation

In this section we demonstrate that the Reward Bases model is closely related to the successor representation [18]. Both models achieve the same performance in tasks with changing rewards for outcomes, but the Reward Bases model has a significantly smaller computational cost.

The successor representation [18] is an alternative model-free RL method that also enables instantaneous generalisation across changing reward functions. It learns a 'successor matrix' based on discounted environmental state occupancies (see Methods Section 4.7 for mathematical details) which can then be combined with the reward function to yield the value function. Given this matrix, it is possible to instantly recompute the value function if the reward function changes.

In the Methods Section 4.7 we show mathematically that the Reward Bases model is closely related to the successor representation and, indeed, can be intuitively thought of as a compressed successor representation tuned only to relevant reward dimensions. Therefore, the Reward Bases and the successor representation have approximately equivalent capabilities. We demonstrate this in a room navigation task introduced in Fig 1.

Our test environment is a $6 \times 6$ grid-world room (Fig 1). The goal is, when started in a random position in the room, to reach a specific type of object as fast as possible. There were three objects of different types that were positioned randomly at the start of the simulation. Agent training was separated into episodes such that whenever the agent reached the valuable object, the episode would end, and the agent would restart in a randomly chosen square of the room. At each moment of time only one object was valuable, and after a certain number of trials objects' desirability was reversed—making one of the three objects valuable while demoting the others.

In Fig 8A, we plot the performance of the models in sample simulation. Fig 8A shows that the successor representation has very similar performance as Reward Bases. Both agents rapidly adapt to reversals but still also require some retraining after reversals likely due to the approximate nature of the learnt value functions or successor matrix given their limited number interactions with the environment. In this task, the Reward Bases and successor representation agents both achieve approximately equal performances across a wide range of parameter settings (Fig 8B and 8C). As expected, both agents perform better than TD learning which needs to completely re-learn the value function after each reversal.

Despite similar performance, the Reward Bases model stores much less information than successor representation. The value bases store $X \times N$ numbers, where $X$ is a number of environmental states and $N$ is the number of reward bases, while the successor matrix stores $X \times X$

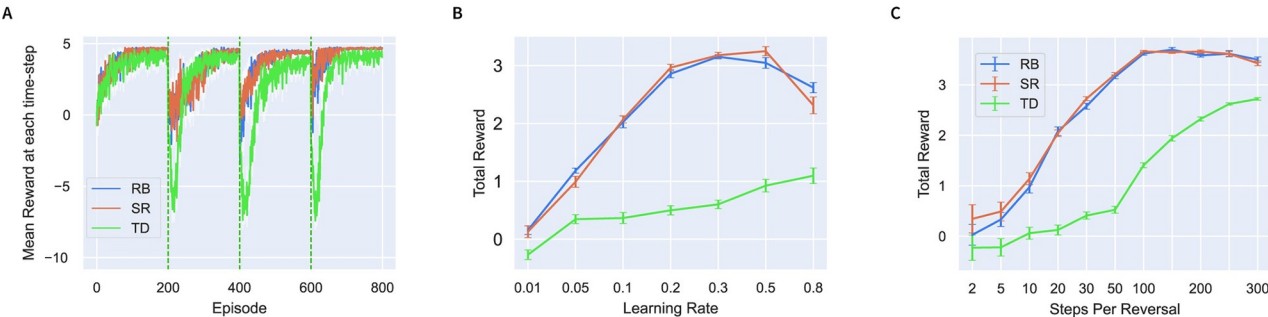

**Fig 8. Comparison of model performance in a room navigation task. A**: Comparison of the reward obtained by the Temporal Difference (TD), the Reward Bases (RB), and the Successor Representation (SR) agents on an example run of the room task. The vertical dashed lines represent the reversals. **B**: Performance obtained by RB, TD, and SR agents over a range of learning rates with 50 episodes between each reversal. **C**: Performance obtained by RB, TD, and SR agents over a range of trials between reversal. A learning rate of 0.1 was used. All error bars represent standard error over 10 runs.

numbers. This difference is already substantial for the small task presented here, where with three reward bases, the number of values stored by the Reward Bases algorithm was $36 \times 3 = 108$ while the successor matrix required $36 \times 36 = 1296$. This advantage will generally only increase with environmental state-size, because the memory cost of the Reward Bases model scales linearly with the environmental state-size, while for the successor representation it scales quadratically.

## 3 Discussion

In this paper, we have proposed a novel mechanism for instant generalisation to changing reward functions in RL tasks which relies solely on model-free learning. The Reward Bases algorithm is straightforward and can be applied to any RL model which estimates values of environmental states or actions. In this section we discuss the relationship of our model to experimental data, other models, and outline possible directions of future work.

### 3.1 Relationship to neural data

We have provided preliminary support for the key experimental prediction arising from our model and demonstrated that a fraction of dopamine neurons encodes the prediction errors associated with one reward type to a larger extent than the error associated with the other type. This observation is consistent with a study showing that separate groups of dopamine neurons increased their tonic firing after intragastric infusion of either food or water [38]. That study also found no correlation across dopamine neurons between responses to food and water, which parallels the lack of correlation in Fig 3F. Our analyses extend the results of this study [38] by suggesting that different dopamine neurons also produce phasic responses after stimuli predicting different types of rewards, and these responses scale with the amount of a particular reward type. Our results are also related with a recent observation that distinct but overlapping populations of dopamine neurons respond to food and social rewards [66].

Our results add support to a general theory that the striatum is composed of neural populations predicting different quantities, and the dopamine neurons projecting to a given population of the striatum compute the error in prediction made by this group of neurons [67]. In the case of Reward Bases model, different populations located in ventral striatum compute different value bases, but other striatal regions may compute different quantities, and consequently corresponding dopamine neurons may encode errors in different predictions. For

example, dopamine neurons in the tail of the striatum were proposed to encode prediction errors specifically for aversive stimuli [68, 69], or action prediction errors [70]. Moreover, dopamine neurons have been shown to encode errors in predictions of sensory features of reward in addition to value [37]. There is also growing evidence that dopamine neurons are sensitive to various aspects of locomotion and kinematic behaviour [36, 71–73] as well as choice behaviour [74, 75] and indeed that this heterogeneity is topographically organised due to spatially organised cortical projections [35, 73].

The existence of striato-dopaminergic modules seems to contradict a common belief that activity of dopamine neurons is transmitted throughout the whole striatum. This belief originates from the observation that dopamine receptors are located outside synapses so they can be only activated by dopamine diffusing through extrasynaptic space [76]. However, it has been pointed out [76] that only a small volume of space around the axonal release site (within $1\mu m$) is likely to reach sufficient dopamine concentration for receptor activation. So although dopamine release is not targeted at single synapses, it has substantial spatial precision. Moreover, despite dopamine neurons being known to have exceptionally large axons, it has been reported [77] that axonal bush of a single dopamine neuron covers on average 2.7% of volume of striatum in the corresponding hemisphere, and this area in concentrated to a particular region of striatum. Hence the existence of modules within the striatum modulated by distinct sets of dopamine neurons is anatomically plausible.

Given the importance of being able to appropriately select actions according to physiological state, similar mechanisms to those considered in this paper may also operate in simpler animals that do not have the basal ganglia. In particular, there is a strong match between the neuroanatomy postulated by the Reward Bases model and that found in the mushroom body of fruit flies (drosophila). This region includes Kenyon cells encoding sensory information, and Mushroom Body Output Neurons (MBON) which are mutually connected with dopamine Neurons (DAN) [78]. A large body of evidence, strongly suggests that the MBONs and DANs are specialised and separated into different zones or compartments which respond to and represent rewarding aspects of specific stimuli, and connectivity between MBONs and DANs is mostly within a given compartment [78–82]. Moreover, experimentally it has been shown that specific DANs respond to specific reinforcement types such as sugar [39, 83], water [21, 40], courtship [41, 84] and aversive stimuli [42, 80, 85, 86], and appear to be instrumental in learning associations based on these specific reinforcement type. Hence if we associate environmental state $x$ with Kenyon cells, value bases $\mathcal{V}_i$ with the MBONs and the prediction errors $\delta_i$ with the DANs, then the circuitry in the mushroom body appears to almost precisely fit the single selectivity implementation of the Reward Bases model.

## 3.2 Comparison with alternative models

Several recent theoretical studies have analysed RL models learning different reward dimensions. Our results are consistent with studies showing that primate and rodent behaviour is well described by models that learn expected reward in multiple dimensions via prediction errors for different reward dimensions [87, 88]. Our paper additionally shows how such learned values may be combined to enable flexible behaviour. Additionally, recent theoretical work has demonstrated that RL agents including separate modules predicting reward in different dimensions have advantages in tasks requiring acquisition of multiple resource types [89].

The Reward Bases model is closely related (but not equivalent) to a multi-objective RL model [48] which also learns separate value functions for different reward dimensions, and combines them while controlling behaviour. In that model, the prediction error for dimension $i$ is computed based on reward scaled by the "drive priority" playing a similar role to our $m_i$, so

in the notation of our paper this prediction error can be written as $\delta_i = m_i r_i(x) + \ldots$. The overall value is computed as a sum of individual value functions, which in our notation can be written as $V(x) = \sum_i V_i(x)$. Despite the similarity of these two equations to Eqs 4 and 5, they are not identical, because in the multi-objective RL model, motivation scales the reward during learning (as it appears in the prediction error), rather than the value function during choice of behaviour (it is not weighting individual values being summed). Although both models generate the same behaviour if motivational drives $m_i$ are constant, they behave very differently after a change in motivation: The Reward Bases model instantly changes the overall value function, while in the multi-objective RL model [48] behavioural change would only follow after experiencing the reward in order to update the value. Consequently, that model cannot capture instant salt revaluation seen in salt-deprivation before any interactions with the salt [10], or instant changes in dopamine responses after devaluation seen before receiving devalued reward [31], while we demonstrated that the Reward Bases model can reproduce these data.

The Reward Bases model is related with the models of homeostatic regulation [44–46], which also consider multiple dimensions of rewards. While the Reward Bases model defines the reward as the linear summation of elementary rewards scaled by their motivational needs, the homeostatic RL defines the reward as a reduction in drive and defines drive as a nonlinear combination of different dimensions of the distance between the current and desired physiological state. Consequently, homeostatic RL can account for multiple experimentally observed phenomena, for example that deprivation of one need reduces the rewarding value of other outcomes [45]. The homeostatic RL model has also been applied to explain behaviour in tasks where animals were repeatedly offered choice between salt solutions, and to explain dopamine responses to salt infusion [90]. However, in these models, the agent learns a single overall value function, rather than separate value bases associated with individual reward dimensions. Therefore, when the motivation changes, these models require experiencing reward to change the overall value function (analogously as the multi-objective RL model discussed above). The contribution of the Reward Bases model is that it describes how value can change without experiencing reward, and hence can capture the data on instant changes in behaviour and dopamine release due to changes in motivation.

The Reward Bases model is also related to a model of RL in multidimensional environment [91]. This study demonstrated that if participants are asked to learn reward probability associated with stimuli described by multiple features, their behaviour is best described by a model assuming that they learn values associated with the individual features, and compute the value of a stimulus as a sum of values of individual features. Although the formulation of this model is different (e.g., it only employs a single prediction error), this study suggests that decomposing value into multiple dimensions is a general feature of biological RL.

The Reward Bases model is also related to a feature specific prediction error model [92]. It explains the diversity of responses of dopamine neurons, by assuming that, due to anatomical constraints, individual dopamine neurons receive inputs from striatal neurons representing value of just a subset of features environmental states. So in this respect that model is similar to Reward Bases. However, that model assumes that as a population, dopamine neurons encode the standard prediction error, and this single prediction error is used to update a single value function represented in the striatum, hence it differs from the Reward Bases which assumes the basal ganglia learn multiple value functions and flexibly combine them according to the physiological state.

It is useful to clarify the distinction between the Reward Bases model and distributional RL [57]. That model assumes that the value function is represented by the probability distribution of expected reward. In that model separate modules learn different percentiles of reward distribution and the heterogeneity of dopamine neurons arises as they encode prediction error

associated with different percentiles of the distribution. By contrast, the Reward Bases model proposes heterogeneity in dopamine neurons in another orthogonal dimension—that of reward type. Since distributional RL does not learn reward associated with different reward dimensions, it cannot explain changes in behaviour and dopamine response after change in motivation. Nevertheless, it is possible to straightforwardly combine distributional RL and the Reward Bases models by estimating a distributional value estimate for each value basis in parallel.

Our results demonstrate that behavioural flexibility in the face of changing reward functions does not require model-based methods as argued before [17] but instead can be handled in a purely model-free manner using a straightforward extension to the classical TD learning. We do not claim that the mammalian brain does not employ model-based RL in tasks involving changes in desirability of outcomes [10, 31], but rather that in addition to model-based RL, model-free processes can also underlie performance on these tasks. We show that model-free TD learning, which is usually considered as inflexible, can be extended to adapt to changing reward functions, and we are proposing a mathematical model and neural implementation of how this flexibility can be achieved in the evolutionarily old structures of the basal ganglia thought to support model-free RL.

As a comparable model-free method for instantly generalising to different reward functions, it is worthwhile to perform a detailed side-by-side comparison of the properties of the successor representation [18] and the Reward Bases model. On a computational level, as discussed in detail in Results, the Reward Bases model achieves the same performance as successor representation, but is much more memory efficient, because it does not require storing the successor matrix. The advantage of the successor representation is that zero-shot reward revaluation can be achieved for *any* reward function. The Reward Bases method requires that the reward function be expressable as a linear combination of reward basis vectors (which themselves can be nonlinear functions of the environmental state). Secondly, the Reward Bases model assumes that these reward basis vectors can be specified beforehand, ideally at the beginning of the task. This means that successor methods alone can handle tasks where the reward function can change in truly arbitrary and unexpected ways over the course of a task. Reward Bases models, on the other hand, has an advantage where the space of potential reward function changes occurs in a relatively small linear subspace of the total reward space which can thus be represented well with a smaller set of basis vectors than the full dimensionality of the environmental state-space. While thus being slightly less generalisable in theory, for biological organisms, these conditions may well hold in practice such that while the reward function changes often, it typically only changes over a relatively well-known range such that a relatively small set of reward bases (relative to the total dimensionality of the world) suffice to cover the reward revaluations that actually occur. For instance, an animal may know that its reward function often changes as a function of its level of satiety, or its level of tiredness and thus these can form natural reward bases.

Another possibility for extending model-free TD learning methods to handle varying reward functions is to simply assume that $x$ includes both environmental and physiological state, and use RL to estimate the value function of such combined state. This method, however, suffers from three substantial disadvantages compared to the Reward Bases approach described in this paper. Firstly, such a model does not incorporate any prior knowledge on how the physiological state affects the value function, so it would need to learn this dependence from training data. Hence training of such a model would take much longer than Reward Bases which already assumes the dependence (Eq 4). Secondly, it entails a substantial increase in the ultimate size of the state-space which hampers the generalizability and sample-efficiency of the resulting algorithm, and is especially acute in the presence of continuously-valued

physiological states which would require the animal to essentially learn its value functions from scratch for every possible setting of the physiological state. Continuously varying physiological states are handled naturally, however, in the Reward Bases model by just using the weighting coefficients. Thirdly, this approach of extending the state-space is not guaranteed to achieve the zero-shot generalisation that the Reward Bases scheme is capable of. This is because it uses standard TD learning which can only associate outcomes with a novel physiological state by directly experiencing those outcomes. For instance, in the salt-deprivation experiments [10], the combined state method may not exhibit instant generalisation but may have to experience several positive pairings of the salt with the salt-deprived physiological state to allow for this generalisation—while in the original experiment the rats were specifically never previously salt-deprived in their lives to prevent this possible association developing outside the experimental paradigm.

The network implementation of the Reward Bases model is based on work [61] pointing out that the striato-dopaminergic circuit is a remarkable example of the brain structure whose known anatomical and physiological properties match those required to implement the feedback alignment algorithm for training of multi-layer neural networks [64]. In particular, the dopamine neurons, which encode the errors in predictions made by the network, project back to the striatum and modulate the plasticity of hidden-layer cortico-striatal weights [8]. In the previous model [61] the dopamine neurons encoded prediction errors associated with just a single dimension of the target. This paper thus extends the previous work and shows that basal ganglia can learn through feedback alignment also when dopamine neurons encode mixtures of prediction errors. While the previous work [61] demonstrated the effectiveness of feedback alignment for training of a model of a motor part of the striato-dopaminergic system, this paper shows that this algorithm is also effective for training the value-based part of the system, suggesting that feedback alignment may be a general principle employed by different parts of the striato-dopaminergic system. If neurons encoding mixtures of prediction errors in other striatal areas are observed in future studies, an analogous model could be applied to explain how the striatum can learn based on such mixed teaching signals.

### 3.3 Generalisations of the model and future work

While introducing the model (Eq 5) we assumed that the temporal discounting $\gamma$ of all the components of the reward is the same. However, this may not be the case, e.g., the temporal discounting for money in humans is much slower than for food or water. Nevertheless, the model could be easily extended to include separate discount factors $\gamma_i$ for different reward dimensions. For such an extended model it would not be possible to show its equivalence to the standard TD learning when weighting coefficients are equal across dimensions (Eq 6), but it would still learn separate value bases. Since experiments re-analysed in this paper did not investigate temporal discounting, such extended model is equally consistent with these experimental data as the models described earlier.

In this paper, we have simply assumed that the motivational drives $m_i$ are known to the agent, but the brain would have to determine their correct values. One way to bring an organism towards a homeostatic set-point is to set the motivation $m_i$ for a specific homeostatic variable proportional to the difference between the optimal set-point of that variable and its current level [60]. So for example, in the case of the salt experiment, the resulting motivation weighting coefficient would be negative when the salt level is higher than optimal, and positive after salt is depleted. Animals possessing the ability to compute motivation weights in this way could thus value salt when depleted even if they never experience its depletion before. Such a model is closely related to drive reduction theory [44, 93, 94] which fits closely with notions of

homeostasis and allostasis in biology. Due to the importance of the computation of this difference between optimal and current physiological levels, it could be hardwired in subcortical regions of the brain such as the hypothalamus. Beyond simple homeostatic set-point controllers, it is possible that more complex controllers also exist to achieve more fine-grained and context sensitive control over more abstract reward types such as monetary rewards. It is also plausible that such controllers would also learn the optimal motivation weighting coefficients for given contexts.

In this paper, we have assumed that a set of reward bases which can correctly reconstruct the reward function is defined a priori. It is likely that there is a set of 'hardcoded' reward bases originating in the hypothalamus and midbrain which directly code for primary rewards. These reward bases may include multiple dimensions of nutrients such as proteins and carbohydrates, because animals are able to balance their intake [95]. They may also include water, salt, sex, as well as other survival-relevant quantities such as social rewards or energetic/metabolic cost. In drosophila at least 3 distinct dimensions of positive reinforcement are encoded by around 100 dopamine neurons [80], so it is plausible that mammals, which have orders of magnitude more dopamine neurons, may represent more dimensions. Additionally the brain may *learn* a good set of reward bases to be able to generalise across the kinds of reward revaluations that happen often in its environment. Computationally learning such bases could be achieved through finding the representations that are optimal for satisfaction of animal's needs [96].

Our study provides initial evidence for learning value in multiple reward dimensions in the basal ganglia based on the limited available experimental data. Thus, future studies utilising a larger number of animals would be necessary to confirm if individual dopamine neurons are selective for value coding in different reward dimensions. Future studies are also needed to identify which dimensions of reward are represented by the dopamine neurons, whether neurons encoding specific dimensions are topographically organised, and if dopamine neurons can learn to represent novel reward dimensions.

## 4 Methods

### 4.1 Temporal difference learning

In multiple simulations we included the TD learning model for comparison with the Reward Bases model. In the TD model, the value function is learned by directly updating the estimated value of the current environmental state [97],

$$\Delta \mathcal{V}(x) = \alpha \delta(x) \tag{10}$$

$$\delta(x) = r(x) + \gamma \mathcal{V}(x') - \mathcal{V}(x) \tag{11}$$

This learning rule is analogous to that in the Reward Bases model, but is based on the total reward $r(x)$ rather than reward bases.

### 4.2 Analysis of activity of individual dopamine neurons

For a full description of the paradigm and data acquisition, please refer to the original paper [43]. The data from this study [43] consists of series of spike-times for each neuron for each trial and the associated condition. There were five conditions corresponding to the monkey being presented with 1.5g banana, 0.3g banana, 0.9ml juice, 0.5ml juice, and 0.2ml juice.

To obtain the subjective values plotted in Fig 3B, we read off the values from the subjective value plot of Figure 4 in the original study [43]. We then shifted the obtained values by +0.5 so

as to move the range of subjective values to lie between 0 and 1. This has no impact on the relative ranking or differences between conditions but makes the resulting regression coefficients more interpretable.

In Fig 3C, 3D and 3E, we plot the average firing rate over the trials of each condition for sample neurons. To obtain a smoothed number of spikes, we counted the number of spikes within a set of overlapping windows, each 200ms long and starting every 50ms. To convert from the raw number of spikes in the window to an average firing rate, we simply divided the number of spikes by the window size (i.e. by 0.2s). We plot the average firing rate of the neuron in each condition. We aligned the timestamps of each trial so that the condition cue was presented at time 0. In the accompanying histograms, we show the distribution across trials of neural firing rates within a window of 150–500ms after stimulus onset for each condition. The same window was used in the analysis of neural data in the original study [43], so we use it as well for consistency. The firing rates are obtained as before by dividing the number of spikes within the window by the window size.

To quantify the extent to which different neurons encode prediction errors associated with juice and banana, we fitted a regression model to each neuron, which predicts the total number of spikes $S$ in a window of 150–500ms after stimulus onset, based on a juice subjective value and a food subjective value regressors:

$$S = \beta_1^k r_1(x) + \beta_2^k r_2(x) + \beta_0^k + \omega \tag{12}$$

where $r_1$ and $r_2$ are the reward bases for the banana and juice regressors, $\beta_1^k$ and $\beta_2^k$ are the regression coefficients of dopamine neuron $k$ for each reward basis while $\beta_0^k$ is the intercept term and $\omega$ represents the noise term. The juice reward basis here assigned the subjective values associated with the juice for the juice conditions and 0 for the food conditions. Conversely, the banana reward basis here assigns the subjective values associated with the banana for the banana conditions and 0 otherwise. For instance, we can formally define the juice reward basis $r_1$ for condition $x$ as,

$$r_1(x) = \begin{cases} R(x), & \text{if } x \in \text{Juice\_Conditions} \\ 0, & \text{otherwise} \end{cases} \tag{13}$$

The subjective values $R(x)$ are those taken from [43] and replotted in Fig 3B. Note that if we used regressors equal to prediction errors rather than $r_i$, the resulting coefficients $\beta_i^k$ would be identical, because the prediction errors just differ from $r_i$ by a constant (expected reward) and these constants are incorporated into $\beta_0^k$ in our regression. The resulting regression coefficients are plotted in Fig 4A.

To statistically quantify if individual dopamine neurons are selective for a particular reward type, we employed a regression analysis. For each neuron we fitted a set of models (listed in Fig 4B) predicting the number of spikes emitted in each trial within a window of 150–500ms after stimulus onset, based on value and identity regressors. The value regressor on a given trial was assigned to the subjective value of reward on that trial found in the original study [43] and plotted in Fig 3B. The identity regressor was coded such that +1 indicated a juice trial and −1 indicated a banana trial.

$$I(x) = \begin{cases} 1, & \text{if juice trial} \\ -1, & \text{if banana trial} \end{cases} \tag{14}$$

The winning model can be described mathematically as:

$$S = b_1^k R(x) + b_2^k I(x) R(x) + b_0^k + \omega \tag{15}$$

where $S$ is the number of spikes within the window, $R(x)$ is the value regressor, $I(x)$ is the identity regressor, $b_1^k$ and $b_2^k$ are the regression coefficients for neuron $k$, $b_0^k$ is the intercept term and $\omega$ is the noise term in the linear model. Fig 4C plots the regression coefficients $b_1^k$ and $b_2^k$ for each neuron $k$.

We now show that the regression model of Eq 15 is equivalent to that of Eq 12. We first note that there exist relationships between regressors in the two models: $r_1(x) = R(x)(I(x) + 1)/2$ and $r_2(x) = R(x)(-I(x) + 1)/2$. For example, if $x$ is one of the stimuli predicting juice, then $I(x) = 1$, so $R(x)(I(x) + 1)/2 = R(x) = r_1(x)$. It is insightful to substitute these relationships into Eq 12:

$$S = \beta_1^k R(x)(I(x) + 1)/2 + \beta_2^k R(x)(-I(x) + 1)/2 + \beta_0^k + \omega \tag{16}$$

$$= (\beta_1^k + \beta_2^k)/2\ R(x) + (\beta_1^k - \beta_2^k)/2\ R(x)I(x) + \beta_0^k + \omega \tag{17}$$

One can observe that Eq 12 can be rewritten as a regression with respect to $R(x)$ and $I(x)$ $R(x)$, and there exist the following relationships between the coefficients in the two models: $b_1^k = (\beta_1^k + \beta_2^k)/2$ and $b_2^k = (\beta_1^k - \beta_2^k)/2$.

## 4.3 Model with state-modulated prediction errors

Reward Bases model with state-dependent prediction error employs a 'modulated prediction error' $\tilde{\delta}_i$,

$$\tilde{\delta}_i = m_i \delta_i = m_i r_i(x) + \gamma m_i \mathcal{V}_i(x') - m_i \mathcal{V}_i(x) \tag{18}$$

The TD update can then be derived as a gradient descent on the squared prediction errors,

$$L_i = \frac{1}{2} \tilde{\delta}_i^2 \tag{19}$$

$$\Delta \mathcal{V}_i(x) = -\alpha \frac{\partial L_i}{\partial \mathcal{V}_i(x)} = \alpha m_i \tilde{\delta}_i \tag{20}$$

This results in a modified TD learning rule for the value bases which effectively defines an adaptive learning rate schedule where the learning rate depends on the physiological state. Although slightly complicating the algorithm, this approach has potential advantages for the brain. It provides a natural scaling of the learning rate with physiological stress, so that the learning rate is higher when the physiological state is more perturbed and hence the $m_i$ weightings are higher. This has clear advantages since it is important to learn fast in such cases. On the other hand, having a reduced (or no) learning rate in the case of satiation may also be beneficial. Reducing the learning rate may reduce the metabolic cost of making the updates, since less synaptic plasticity is required and the brain has been heavily optimised by evolution to minimise energy expenditure [98].

Please note that adding the state-modulated prediction error to the standard TD model, would on its own not be able to explain dopaminergic activity or behaviour seen on the first trial after changes in reward values [10, 31], because the TD model would still require an interaction with the environment for at least one trial to update the values. To explain the

dopamine activity on the first trial after changes in reward values [31], we added the state-modulated prediction error to the Reward Bases model.

## 4.4 Mixed selectivity model

We constructed a model in which each dopamine neuron encoded a weighted sum of prediction errors, and demonstrated that this model can learn correct value bases in the task of Lak et al. [43]. The architecture of the model and notation used are shown in Fig 5A. We index variables associated with dopamine and striatal neurons with superscripts, to highlight that they encode prediction errors and value bases in distributed fashion, and to distinguish from the dimension-specific prediction errors and value bases indexed with subscripts in previous variants of the Reward Bases models. Each dopamine neuron in the model receives input from reward areas and inhibition from the striatum:

$$\tilde{\delta}^k = \sum_{i=1}^{2} w_{k,i}^{R \to D} m_i r_i - \sum_{i=1}^{M} w_{k,i}^{S \to D} s^i \tag{21}$$

The inhibition from the striatum provides information on the expected value. This prediction error term contains no future value term corresponding to $\gamma \mathcal{V}_i(x')$ since our task lasted for just a single step. We assume that dopamine neurons in addition to coding the above prediction error (e.g. in the phasic activity) also encode the corresponding motivational drives (e.g. in their tonic activity):

$$D^k = \sum_{i=1}^{2} w_{k,i}^{R \to D} m_i \tag{22}$$

The activity of striatal neurons depends on their input from the cortex:

$$s^k = g^k \sum_{i=1}^{5} w_{k,i}^{C \to S} x_i \tag{23}$$

In the above equation $g^k$ is the gain of striatal neuron $k$ which is determined by motivational drives encoded by dopamine neurons projecting to striatal neuron $k$:

$$g^k = \sum_{i=1}^{N} w_{k,i}^{D \to S} D^i \tag{24}$$

Setting the gain of striatal neurons in the above way enables the value bases encoded by striatum to be weighted by motivational drives, as required by the Reward Bases model. The value function is encoded in the model by the total output from the striatum received by all dopamine neurons, defined as:

$$S^{out} = \sum_{k=1}^{N} \sum_{i=1}^{M} w_{k,i}^{S \to D} s^i \tag{25}$$

For this striatal output to be equal to the value function, the weights from reward areas to dopamine neurons need to be normalised so that each reward basis is equally represented by

the dopamine neurons:

$$\sum_{k=1}^{N} w_{k,i}^{R \to D} = 1 \tag{26}$$

We now demonstrate that if all prediction errors are equal to 0, then the total striatal output is equal to the value function. If all $\tilde{\delta}^k = 0$, then from Eqs 25 and 25:

$$S^{out} = \sum_{k=1}^{N} \sum_{i=1}^{2} w_{k,i}^{R \to D} m_i r_i \tag{27}$$

Using Eq 26, we see that the striatal output simplifies to:

$$S^{out} = \sum_{i=1}^{2} m_i r_i \tag{28}$$

We observe that the striatal output is indeed equal to the sum of the expected rewards weighted by their corresponding motivational drives. Since encoding of the value function by striatal output is enabled by prediction errors being equal to 0, the synaptic weights are modified to minimise the prediction errors. Hence we define the loss function:

$$L = \frac{1}{2} \sum_{k=1}^{N} (\tilde{\delta}^k)^2 \tag{29}$$

The striato-dopaminergic weights are found by the gradient descent on the loss:

$$\Delta w_{k,i}^{S \to D} = -\alpha \frac{\partial L}{\partial w_{k,i}^{S \to D}} = \alpha \tilde{\delta}^k s^i \tag{30}$$

The resulting update of a particular weight is equal to the product of the activity of the post-synaptic neuron $\tilde{\delta}^k$ and the pre-synaptic neuron $s^i$, so it corresponds to Hebbian plasticity. The gradient of the loss over cortico-striatal weights is equal to:

$$-\frac{\partial L}{\partial w_{i,j}^{C \to S}} = \sum_{k=1}^{N} w_{k,i}^{S \to D} \tilde{\delta}^k g^i x_j \tag{31}$$

Modifying cortico-striatal weight according to the above rule would be biologically unrealistic because it assumes scaling of dopamine inputs by striato-dopaminergic weights rather than reciprocal dopamine-striatal weights through which the dopamine signals are actually transmitted. Therefore, we follow the models employing feedback alignment [61, 64], and modify the cortico-striatal weights according to:

$$\Delta w_{i,j}^{C \to S} = \alpha \sum_{k=1}^{N} w_{i,k}^{D \to S} \tilde{\delta}^k g^i x_j \tag{32}$$

Note that the above learning rule corresponds to local synaptic plasticity, because $\sum_{k=1}^{N} w_{i,k}^{D \to S} \tilde{\delta}^k$ is the total dopaminergic prediction error received by the postsynaptic neuron $i$, $g^i$ is the gain of the post-synaptic neuron, and $x_j$ is the activity of the pre-synaptic neuron.

We simulated a network with 5 cortical neurons, as there were 5 stimuli in the study of Lak et al. [43], and 2 reward neurons as there were 2 reward types in this study. The network also included $M = 50$ striatal neurons and $N = 10$ dopamine neurons. All weights were initialised according to Xavier uniform initialisation [99], and the weights from reward to dopamine

neurons were additionally normalised according to Eq 26. At the start of each simulated trial, a stimulus was randomly chosen, so the corresponding cortical neuron was set to $x_j = 1$, while other cortical neurons were set to 0. The reward basis corresponding to the type of reward presented by the stimulus was set to the subjective value (Fig 3B), while the other reward basis was set to 0. The motivational drives were chosen randomly from uniform distribution [0, 1]. The activities of striatal and dopamine neurons were computed, and then the cortico-striatal and striato-dopaminergic weights were updated with learning rate $\alpha = 0.05$. This training procedure is summarised in Algorithm 1. To evaluate a stimulus based on cortical inputs $x_j$ and motivational drives $m_i$, the following variables were evaluated in turn: $D^k$ (Eq 22), $g^k$ (Eq 24), $s^k$ (Eq 23), $S^{out}$ (Eq 25).

**Algorithm 1:** Training of the mixed selectivity model.

```
input: juice rewards for stimuli R₁ = [1, 0.5, 0.1, 0, 0], banana
       rewards for stimuli R₂ = [0, 0, 0, 0.7, 0.05]
output: synaptic weights
Set learning rate α = 0.05
Set number of training iterations ITER = 100000
Initialise weights wᶜ→ˢ, wˢ→ᴰ, wᴰ→ˢ, wᴿ→ᴰ with uniform Xavier
Normalise weights wᴿ → ᴰ (Eq 26)
for it = 1 to ITER do
  stimulus = random from {1, .., 5}
  x_stimulus = 1, x_{j≠stimulus} = 0
  r₁ = R₁[stimulus], r₂ = R₂[stimulus]
  m_i = random from [0, 1]
  Compute Dᵏ (Eq 22)
  Compute gᵏ (Eq 24)
  Compute sᵏ (Eq 23)
  Compute δ̃ᵏ (Eq 21)
  Update w_{k,i}^{S→D} (Eq 30)
  Update w_{i,j}^{C→S} (Eq 32)
```

The total striatal output $S^{out}$ is sent to dopamine neurons in the model, while to be useful for other brain systems, it needs to be transmitted also to them. There are several possibilities in which this transmission may happen, for example, the striatal neurons could send collateral projections to other structures, or the striatal neurons could send their projections to intermediate neurons (e.g. in the output nuclei) that could project both to dopamine neurons and thalamus, as in a previous model [61].

A summary of different variants of Reward Bases models is given in Table 1.

## 4.5 Modelling the dependence of dopaminergic response on physiological state

The Reward Bases agent was trained in a simulated version of the task paradigm used in the original study [31] and graphically described in Fig 6A. The agent maintained two reward and

**Table 1. Summary of variants of the Reward Bases model.**

| Model | Prediction error | Value update | Evaluation |
|---|---|---|---|
| Reward Bases | $\delta_i = r_i(x) + \gamma \mathcal{V}_i(x') - \mathcal{V}_i(x)$ | $\Delta \mathcal{V}_i(x) = \alpha \delta_i$ | $\mathcal{V}(x) = \sum_i m_i \mathcal{V}_i(x)$ |
| State-modulated | $\tilde{\delta}_i = m_i \delta_i$ | $\Delta \mathcal{V}_i(x) = \alpha m_i \tilde{\delta}_i$ | $\mathcal{V}(x) = \sum_i m_i \mathcal{V}_i(x)$ |
| Mixed selectivity | $\tilde{\delta}^k = \sum_i w_{k,i}^{R \to D} m_i r_i - \sum_i w_{k,i}^{S \to D} s_i$ | $\Delta w_{i,j}^{C \to S} = \alpha \sum_k w_{i,k}^{D \to S} \tilde{\delta}^k g^i x_j$ $\Delta w_{k,i}^{S \to D} = \alpha \tilde{\delta}^k s^k$ | $\mathcal{V}(x) = \sum_k \sum_i w_{k,i}^{S \to D} s^i$ $s^k = g^k \sum_i w_{k,i}^{C \to S} x_i$ $g^k = \sum_i w_{k,i}^{D \to S} D^i$ $D^k = \sum_i w_{k,i}^{R \to D} m_i$ |

value bases $\mathcal{V}_i(x)$ over the two environmental states of the experiment $x_k$ corresponding to pressing the two levers. On each trial the agent was choosing between two options, with probability of selecting option $k$ equal to:

$$P_k = \frac{e^{\mathcal{V}(x_k)}}{e^{\mathcal{V}(x_1)} + e^{\mathcal{V}(x_2)}} \tag{33}$$

In the above equation, $\mathcal{V}(x_k)$ was computed from Eq 5 with $m_i = 1$. After choice, there was a 80% chance of getting corresponding reward type, a 15% percent chance of switching to the other reward type (SWITCH condition) and a 5% percent chance of getting 4 outcome units (MORE condition). Consequently both value bases for chosen option $k$ were modified according to

$$\Delta \mathcal{V}_i(x_k) = \alpha(r_i(x_k) - \mathcal{V}_i(x_k)) \tag{34}$$

In the above equation, the discounted future reward was not included, because no further reward was expected, and learning rate was set to $\alpha = 0.1$. Simulations of learning were repeated 3 times with unique seeds of random number generator, and within each repetition the value functions were learned over 500 trials.

The learned value bases were used to simulate dopaminergic responses in all experimental conditions in Fig 6C and 6D. Due to similarity of dopaminergic responses to food and water when these reward were valued (or devalued) seen in Fig 6B, we summarised these data by the dopamine concentration averaged across valued (or devalued) trials, as illustrated by blue shading from Fig 6B to 6D. The dopaminergic responses were simulated as the sum of the weighted prediction errors, in accordance with the modulated prediction error model presented in Section 4.3. That is, we identify dopaminergic response in environmental state $x$ as

$$\sum_i \tilde{\delta}_i(x) \tag{35}$$

Without the loss of generality, we assumed that food was a valued option, while sucrose was devalued. Thus for example, the response to valued lever (Fig 6C) was computed by substituting Eq 18 into Eq 35

$$\gamma m_{\text{food}} \mathcal{V}_{\text{food}}(x_{\text{food}}) + \gamma m_{\text{sucrose}} \mathcal{V}_{\text{sucrose}}(x_{\text{food}}) \tag{36}$$

where $x_{\text{food}}$ denotes the lever commonly associated with food. For simplicity we set $\gamma = 1$ in the simulations. The response to devalued lever was computed analogously. Similarly, the response to the expected valued reward (Fig 6D) was computed as

$$m_{\text{food}}(1 - \mathcal{V}_{\text{food}}(x_{\text{food}})) + m_{\text{sucrose}}(0 - \mathcal{V}_{\text{sucrose}}(x_{\text{food}})) \tag{37}$$

Responses to other outcomes were computed analogously.

The $m_{\text{food}}$ and the $m_{\text{sucrose}}$ are two free parameters. We fitted a single set of $m_{\text{food}}$ and $m_{\text{sucrose}}$ to all experimental values in Fig 6C and 6D, which is in total 8 points to fit (2 in Fig 6C and 6 in Fig 6D). The fitting was done by minimising a least squares error between the model prediction and the corresponding experimental value. To identify the best fitting free parameters we re-parameterised them as $m_{\text{food}} = KR$ and $m_{\text{sucrose}} = K$, where $R$ is a the ratio of weights, and $K$ is a scaling parameter. We sought $R$ through grid searched between 1 and 10, with an interval of 0.1; then, a coefficient $K$ was solved analytically for each $R$ in the grid search to minimise a squared error between the model prediction and the corresponding experimental value. The above procedure of optimising $R$ and $K$ is equivalent to optimising $m_{\text{food}}$ and $m_{\text{sucrose}}$. The advantage is that if optimising $m_{\text{food}}$ and $m_{\text{sucrose}}$, it has to be done via function

evaluation (e.g., grid search on both of them); but while optimising $R$ and $K$, only $R$ need to be optimised by function evaluation (in our case, grid search), but $K$ can be solved analytically.

The experimental values were extracted from dopamine signals 2 seconds after the corresponding events as described in the caption of Fig 6. The extraction was done via a online digital plot extractor at https://apps.automeris.io/wpd/, our extraction can be loaded to this online tool via "loading project", and the project files are located in directory Simulations/dopamine_release/data_extraction/ of the repository associated with this paper, described in Data Availability Statement.

## 4.6 Simulation of dead sea-salt experiment

In the simulated version of the experiment [10] presented in Fig 7, the agents were exposed to the lever associated with the salt or the lever associated with the juice at random over 100 trials. If the salt lever was presented, the TD agent received a reward of $-1$ while if the juice lever was presented the agent received a value of $+1$. The TD agent learnt a value function with 2 environmental states—juice and salt. The Reward Bases agent maintained separate salt and juice reward bases with the juice reward basis returning $+1$ for juice and 0 for salt and vice-versa for the salt basis. Reward Bases agent learned with state-modulated prediction errors. To match the rewards received, during training we set $m_{\text{salt}} = -1$ and $m_{\text{juice}} = 1$. The agents were trained with a learning rate of $\alpha = 0.1$. Since there are no multi-step dependencies in this task, we trained with $\gamma = 0$.

To match the value functions of our agents to behavioural data, we assumed that the degree of appetitive approach and 'liking' behaviour plotted in the original experiment varied linearly to the value of the stimulus the animal had learned. That is, we simulated the simple equation,

$$\text{approaches}(x) = \beta_1 \mathcal{V}(x) + \beta_0 \tag{38}$$

Regression parameters $\beta_j$ were fitted for each model to data from the normal (non-depleted) condition. During testing in the deprivation conditions, the value functions remained the same for TD learning, while they were recomputed for Reward Bases with $m_{\text{salt}} = 0.5$, as this value resulted in quantitative match with experimental data.

## 4.7 Successor representation

Let us first introduce the notation required to describe the successor representation. Let $\mathbf{r}$ denotes a vector containing expected instantaneous rewards of all environmental states, i.e. a vector of a length equal to the number of possible environmental states, where each entry is equal to expected instantaneous reward for the corresponding environmental state. Analogously, let $\mathbf{V}$ denote the vector of values of all environmental states. Furthermore, let $\mathcal{T}$ denote matrix with environmental state transition probabilities, where entry $\mathcal{T}_{y,x}$ denotes the probability of agent transitioning from environmental state $x$ to $y$ under the current policy. Using this notation, the definition of the value function from Eq 1 can be written as follows.

$$\mathbf{V} = \mathbf{r} + \gamma \mathcal{T} \mathbf{r} + \gamma^2 \mathcal{T}^2 \mathbf{r} \cdots \tag{39}$$

By a simple rebracketing, we can express this as,

$$\mathbf{V} = (I + \gamma \mathcal{T} + \gamma^2 \mathcal{T}^2 \cdots) \mathbf{r} \tag{40}$$

We can then separate out the matrix $(I + \gamma \mathcal{T} + \gamma^2 \mathcal{T}^2 \ldots)$ and call it $\mathcal{M}$, the successor matrix giving us the following equation for the value function,

$$\mathcal{V}(x) = \sum_y \mathcal{M}_{x,y} \mathbf{r}_y \tag{41}$$

It is therefore clear that if we know $\mathcal{M}$ and have it stored, then, given any change to the reward function $\mathbf{r}$, it is trivial to recompute the correct value function. This allows for instantaneous reward revaluation since the value function can be recomputed so easily.

The successor matrix $\mathcal{M}$ can be learned from agent's experience. Each time an agent transitions from environmental state $x$ to $x'$, all entries in row $x$ of matrix $\mathcal{M}$ are updated,

$$\Delta \mathcal{M}_{x,:} = \alpha(\mathbf{x} + \gamma \mathcal{M}_{x',:} - \mathcal{M}_{x,:}) \tag{42}$$

where $\mathbf{x}$ is one-hot environmental state vector which signifies the environmental state of the agent before the transition (i.e. its entry corresponding to $x$ is equal to 1 while other entries to 0), and $\mathcal{M}_{x,:}$ denotes the row $x$ of matrix $\mathcal{M}$.

It is possible to show mathematically that the Reward Bases model is closely related to the successor representation and, indeed, can be intuitively thought of as a compressed successor representation with rewards tuned only to relevant dimensions. Recall that according to Eq 40, the value function can be expressed as $\mathbf{V} = \mathcal{M}\mathbf{r}$, so analogously a value basis can be expressed as

$$\mathbf{V}_i = \mathcal{M}\mathbf{r}_i \tag{43}$$

This can be demonstrated by substituting Eq 2 into Eq 40:

$$\mathbf{V} = \mathcal{M}\mathbf{r} = \mathcal{M}\sum_i m_i \mathbf{r}_i = \sum_i m_i \mathcal{M}\mathbf{r}_i \tag{44}$$

Since the value function also satisfies $\mathbf{V} = \sum_i m_i \mathbf{V}_i$, the condition of Eq 43 must hold.

The relationship of the Reward Bases model with the successor representation (Eq 43) helps explain why the Reward Bases model achieves equivalent performance to the successor representation at a significantly smaller computational cost.

## 4.8 Simulations of the room task

The simulated environment was similar to that in Fig 1, i.e. it consisted of a $6 \times 6$ grid world containing 3 objects. Each of these objects had reward $r_i = 5$ in one of the dimensions, while $r_i = -1$ in the other dimensions. At each trial, the total reward was set to one of the reward bases, and the valued dimension was changed at the reversals.

For all simulated agents, the value function was represented as a flattened vector of $6 \times 6 = 36$ environmental states, which was initialised at 0. For the successor representation agent, the successor matrix was initialised as a $36 \times 36$ matrix of 0s and was updated on each timestep by the successor representation update rule (Eq 42). The successor agent computed its estimated value functions according to Eq 41.

For all agents a learning rate of $\alpha = 0.05$, a discount factor of $\gamma = 0.9$ and a softmax temperature of 1 were used. Actions were selected by random sampling over the softmaxed distribution over actions. We used 500 steps between reversals. Means and standard deviations were obtained over 10 seeds for each agent in Fig 8.

## Supporting information

**S1 Fig. Average firing rates during a trial for all recorded neurons.** The plot shows the firing rate as a function of time within a trial (see Methods Section 4.2), where time 0 corresponds to onset of the stimulus indicating which reward type will be presented. The neurons are ordered by the interaction coefficient of value and identity (i.e., as in Fig 4C).
(TIF)

**S2 Fig. Comparison of the ability of different regression models to describe the activity of dopamine neurons after stimuli predicting rewards.** The graph shows results of an analysis analogous to that in Fig 4B, but here Bayesian Information Criterion (BIC) is measured and reported.
(TIF)

**S3 Fig. Model recovery analysis investigating if the Akaike Information Criterion (AIC) can reliably distinguish between the two models from Fig 4B with formulas: "value + identity:value" and "stimulus".** For each model, we generated 1000 surrogate datasets, each containing surrogate activity of the same number of neurons on the same number of trials as in the experiment. To obtain surrogate data, we generated for each trial a prediction with the two fitted models. When generating the prediction, we also added a random number from a normal distribution with the standard deviation equal to the standard deviation of the residuals obtained when we fitting the model to the data from a given neuron. We then fitted both formulas to the data generated from both models, and summed the AIC score across neurons. We plot the results as a histogram of the difference of AIC score between the fit with formula "stimulus" and the fit with formula "value + identity:value". Different colours indicate the model used to generate the surrogate data. There is little overlap between orange and blue histograms indicating that the AIC can reliably distinguish between the two corresponding models. The difference between the AIC scores computed from real data for the two models in Fig 4B was 16.8. Such or higher difference was obtained only for 0.8% of surrogate datasets generated from "stimulus" model, indicating that it is very unlikely for the actual data to be generated by that model.
(TIF)

**S4 Fig. Stability of the selectivity of dopamine neurons for different reward types.** Here, the same analysis as in Fig 4C is conducted, i.e., the regression model "value + identity: value" is fitted to the data, but this is done separately for different portions of the data—first time based on the first half of the trials in the recording session, and second time using the second half of the trials. The interaction coefficients from the two fittings are plotted in a scatter graph, in which each dot corresponds to a neuron, and its x and y coordinates correspond to the coefficients. As can be seen, there is a significant correlation ($r = 0.69$, $p = 0.001$) between the coefficients from the two periods, indicating that the selectivity of neurons for different reward types is stable. The solid line represents the best-fit linear regression line. The shaded area around the regression line, indicates the 95% confidence interval.
(TIF)

## Acknowledgments

The authors would like to thank Nathaniel Daw and Scott Waddell for discussion and Mycah Banks for her aid in preparing the figures. Simulations in Fig 5 were conducted by R.B. while visiting the Okinawa Institute of Science and Technology (OIST) through the Theoretical Sciences Visiting Program (TSVP).

## Author Contributions

**Conceptualization:** Beren Millidge, Rafal Bogacz.

**Data curation:** Yuhang Song, Armin Lak.

**Formal analysis:** Beren Millidge, Yuhang Song, Armin Lak, Rafal Bogacz.

**Software:** Beren Millidge, Yuhang Song, Rafal Bogacz.

**Visualization:** Yuhang Song.

**Writing – original draft:** Beren Millidge, Rafal Bogacz.

**Writing – review & editing:** Yuhang Song, Armin Lak, Mark E. Walton.

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
