## [Decision Letter · Decision Letter 0]

2 Jun 2024

Dear Prof. Bogacz,

Thank you very much for submitting your manuscript "Reward Bases: A simple mechanism for adaptive acquisition of multiple reward types" for consideration at PLOS Computational Biology.

As with all papers reviewed by the journal, your manuscript was reviewed by members of the editorial board and by several independent reviewers. In light of the reviews (below this email), we would like to invite the resubmission of a significantly-revised version that takes into account the reviewers' comments.

We cannot make any decision about publication until we have seen the revised manuscript and your response to the reviewers' comments. Your revised manuscript is also likely to be sent to reviewers for further evaluation.

Sincerely,

Jonathan Rubin

Academic Editor

PLOS Computational Biology

Lyle Graham

Section Editor

PLOS Computational Biology

Reviewer's Responses to Questions

**Comments to the Authors:**

Reviewer #1: It is often assumed that dopaminergic neurons encode TD-error of some global value function. Millidge et al. suggest that in fact, reward is not a scalar, and different dopaminergic neurons encode different combinations of reward prediction errors, corresponding to different reward modalities. This idea is supported by theoretical modeling and analysis of data from two different experiments.

I find the results interesting, and I have only minor comments:

The main comment relates to prior literature. The theoretical idea is not new, and the authors should discuss in more details the specific theoretical contribution of their work beyond references 44-46.

Another, more minor comment relates to the analysis of the experimental work of reference 43. The authors carefully acknowledge that the number of neurons that could be used for the analysis is very small (19) which makes it difficult to draw conclusions. Given the centrality of this experiment in this manuscript, I suggest that all neurons will be presented as a supplementary figure, and not only the examples of figure 3.

Another, even more minor point is that I find Figure 1 and the accompanying text to be trivial, and hence not very helpful. I recommend shortening the manuscript by removing it.

Finally, there are many references to the Methods section. Given the large number of sections in the Methods section, it would be useful to refer the reader to the particular subsection of the Methods section.

Reviewer #2: This manuscript describes a theoretical model meant to reconsider how animals implement reinforcement learning by examining the influence of states on the value function. Specifically, the authors propose a multi-value model that incorporates states (and potentially other factors) into a distribution of value estimates. The implementation of learning multiple values, where reward value is moderated by a state term for the various states of the agent, is proposed to occur within the circuitry of the basal ganglia. The described model predicts that individual dopamine neurons should encode errors associated with some reward dimensions more than others.

A reanalysis of archival data was conducted on a small dataset obtained from a single NHP in an experiment that, according to the authors, is the only published study where the responses of dopamine neurons were recorded after stimuli predicting distinct types of rewards. The analysis revealed that, in addition to encoding subjective economic value, dopamine neurons encode a gradient of reward dimensions; some neurons respond most to stimuli predicting food rewards, while others respond more to stimuli predicting fluids. Furthermore, the model replicates the generalization to new physiological states observed in dopamine responses and behavior. The authors argue that this model demonstrates how a simple neural circuit can flexibly guide behavior according to an animal’s needs.

Overall, this paper is both elegant and intuitive, and the authors make substantial efforts to validate the model against behavioral and neural data. There are no major critiques regarding the structure of the model, but there are some issues in the validation process that need to be addressed.

MAJOR COMMENTS

1. Data Analysis

The model comparison applied to the regression analysis used to validate the model with the first archival data set ends by selecting a model with a main effect of value and an interaction term for value and stimulus identity. However, this violates the hierarchical principle in regression, which requires all main effect terms for all factors used in the interaction term to be present in the model. Otherwise, the interaction term absorbs variance from the missing main effect term and impacts overall model performance, regardless of the statistical significance of the individual regression coefficients.

This is a major issue because the fact that this model, with the interaction term, was the best fit to the data is used as a validation of the predictions of the Reward Bases model. Based on the AIC values shown in Figure 4A, the stimulus alone model would be the best fit if the value + value*identity model were removed from the comparison set. If I read this correctly, that would not validate the theoretical model the authors have proposed. This seems important.

Additionally, regarding data analysis, it is unclear how the permutation tests were run, whose results are plotted in Figures 3G and 4C. I am unsure as to what I am looking at in these plots and how they relate to the relevant null hypothesis test being performed.

2. Logic Structure

One major, but not critical, issue with the logic structure is that the evaluations of the model seem to start with the most complex cases (representations of dopamine neurons) and then move to the simplest (demonstration of how physiological states impact behavior). This seems odd. Wouldn’t it make sense to start by showing the simple fact that the proposed model can capture a classic behavioral effect (i.e., choice behavior with altered physiological states described in the “Behavior reflects instant revaluation in novel physiological state” section) and then demonstrate that the representation of value in neurons matches the mixed selectivity predicted by the model?

3. Model Comparisons

As I was reading this, I kept thinking about how similar this model appears to be to distributional RL. The authors correctly state that both successor representation and distributional RL are quite similar in structure but different in both form and predictions than the current proposed model (see Discussion). However, the only direct comparison in the paper is done with successor representation (and simple TD learning, which is obviously going to be ineffective). So, I am curious why the authors chose to exclude the other class of learning models most similar to the Reward-Based model.

MINOR COMMENTS

- Pg 1, Abstract: “recorded after stimuli predicting distinct types of rewards” is a sentence fragment. I believe something like “were delivered” should be at the end.

- Pg 2, Line 43: “for controlling behaviour” is a typo. It should be “to control behaviour.”

- Pg 11, Line 268: “coefficient” should be “coefficients.”

- Pg 11, Line 280: “with a dopamine neurons” should be “with dopamine neurons.”

- Pg 11, Lines 280-299: The “Learning with mixed selectivity” section feels very rushed. This needs to be elaborated on a bit more.

- Pg 13, Line 320: “more of food” should be “more food.”

Signed: Timothy Verstynen

Reviewer #3: Review of:

Reward Bases: A simple mechanism for adaptive acquisition of multiple reward types.

In this paper, the authors propose and test a simple extension of the classic TD-learning algorithm meant to explain the ability of animals to quickly adapt their preferences for different reward types according to changing physiological or motivational states. The proposed model, called “Reward Bases”, is grounded in two basic assumptions: 1) reward is multi-dimensional, rather than scalar, and 2) value is computed as a weighted sum (dot product) over the value functions of each reward dimension, with weights reflecting the instantaneous motivational drives of the animal with respect to the corresponding reward type. The authors propose a possible implementation of this learning algorithm in basal ganglia circuitry and show that it can explain several findings related to shifting physiological states in dopamine responses and behavior.

The model is elegant and the paper is overall well written. The assumption driving the model nicely align with recent theoretical perspectives (re)emphasizing the role of intrinsic constructs such as, motivation, drives or goals in explaining how animals represent and adapt to their environment. In particular, the assumption of a multi-dimensional, motivation dependent reward function consistent with recent theoretical and experimental work suggesting stronger coupling than previously thought between descriptive (representational) and normative (evaluational) aspects of goal-directed learning. However, I feel that this point could have been made stronger if the authors disambiguated the multiple meanings of the term “state” used in the paper and made it clearer where, according to their model, the representational aspects end and the evaluative ones begins. Also, the authors provide different interpretations to variants of the model that appear to be mathematically equivalent so it was not fully transparent to me where the formal description of the model ends and its implementation or interpretation in a particular context begins. Finally, I think that the paper would benefit from a more substantial comparison with similar existing modeling approaches.

Major comments:

The use of the term “state” in the paper is somewhat equivocal. For example, in lines 7-8 the authors write that a simple neural circuit “learns the availability of different reward types in different states, and then combines them according to the motivational state to control behavior.” (my emphasis). In line 13 they refer to “cortical states” and in line 23 to rats “placed in a physiological state”. In lines 71-72 they describe the agent’s state in a technical sense of Reinforcement Learning (RL) models, as something that “may correspond to an animal’s location in space or a stimulus presented during an experiment”. It would be useful to delineate and clarify the relationship between the different uses – technical and colloquial – of the term “state”. Furthermore, states in RL are supposed to be summaries of the relevant features of experience for guiding goal-directed learning and behavior. Shouldn’t then motivational or physiological “states” also be considered as part of the RL state? On a related point, the authors mention (lines 50-52) that dopamine neurons encode information about diverse variables (which are not directly reward related) such as movement and sensory features. In other words, dopamine neurons seem to encode state information, not only reward prediction error, as posited by TD learning algorithms, regardless of whether reward is thought of a scalar or multi dimensional. It would be interesting to know how this “state-representational” aspect of dopamine activity can be interpreted in light of their model.

How is the lack of correlation between individual neuron responses to juice and banana (Fig 3G) reconciled with the fact that most neurons exhibited mixed selectivity to both reward types (Fig 5A)?

The simulation demonstrating the physiological state dependent reevaluation aspect of the model felt a bit underwhelming. It shows that adding a state dependent parameter on the reward modulates state dependent approach behavior (by the way, I am not sure how approach behavior is defined, this should be explained in the caption or text explaining figure 7). It would be nice to see a less trivial demonstration of this important aspect of their model.

The authors should include more detailed comparisons between their model and related ones such as feature weighted reinforcement learning, e.g., in Niv et al. 2015, the feature-specific reward prediction error model of Lee et al. 2022 or the multithreaded TDRL model of Takahashi et al 2023. More generally, it might be useful to compare the reward bases model with other theoretical frameworks coupling the representational and evaluative aspects of goal-directed learning, e.g., Amir et al 2023.

Line 620 ff. It would be helpful to motivate the construction of this model in Eq. 13, especially as later it is claimed to be equivalent to that of Eq. 12. If they are equivalent, why not just use one? Also, it might be interesting to compare the linear regression model with more realistic (nonlinear) firing rate models.

The authors include error bars on some of their simulation results (Fig 6C and Fig. 8 B-C). However, wouldn't the size of the error bars simply reduce with the number of simulation runs (3 in Fig. 6 and 10 in Fig. 8), and if so what is the meaning of including the bars (and how was the number of simulation runs chosen)?

Fig. 4A, what would the AIC be for a model with only identity as a regressor?

It would help to provide a comparison of the different model variants in one place. Specifically, the mixed selectivity and model and the physiological state-dependent model are interpreted as different but the way in which the dopaminergic response is fitted to the total prediction error in each one (Eqs. 15 & 23) seems mathematically equivalent, with m_i playing an analogous role to \\beta_i. It would be good to make this equivalence clear, or explain if there is some essential difference between, other than that the parameters are interpreted differently, e.g., as neuronal selectivity in the first case and motivational weight in the second (or something similar).

References:

Niv, Yael, et al. "Reinforcement learning in multidimensional environments relies on attention mechanisms." Journal of Neuroscience 35.21 (2015): 8145-8157.

Lee, Rachel S., et al. "A feature-specific prediction error model explains dopaminergic heterogeneity." bioRxiv (2022): 2022-02.

Takahashi, Yuji K., et al. "Dopaminergic prediction errors in the ventral tegmental area reflect a multithreaded predictive model." Nature Neuroscience 26.5 (2023): 830-839.

Amir, Nadav, et al. "States as goal-directed concepts: an epistemic approach to state-representation learning." arXiv preprint arXiv:2312.02367 (2023).

Minor comments:

In the second to last row of tha abstract, “new physiological state” should be “new physiological states”.

Line 13: “large amount” should be “large amounts”.

Line 43: “for controlling” should be “to control”.

Line 81: “seek” should be “seeks”.

Line 210: the authors claim that “multiple neurons decreased their firing”, however according to figure 3F only two neurons show significantly decreased firing rates.

Lines 245-8: the text refers to grey, blue and yellow bars but the figure only seems to have blue and orange bars.

Line 440: what does it mean “rewards are more distinct”, i.e., in what sense are they more distinct?

Line 488: what do the ellipses in the equation stand for?

Lines 533-534 the description of the extended state-space and how it similarly extends TD learning was not completely clear.

Figure 5B is not so clear. In particular, all horizontal lines are dashed while according to the legend the higher valued ones should be solid.

The description of the experimental paradigm in Fig. 6 was a bit unclear. The text says that animals were presented with a cue indicating which of the levers trigger reward delivery but the figure seems to show two cues and subsequently two levers extended (presumably representing a forced choice trial?).

Some of the references seem to contain some scrambled or omitted text (e.g. [1])

**Have the authors made all data and (if applicable) computational code underlying the findings in their manuscript fully available?**

Reviewer #1: None

Reviewer #2: **No: **There is a statement that all data and code "will be made publicly available" but it was not available at the time of this review.

Reviewer #3: Yes

PLOS authors have the option to publish the peer review history of their article (what does this mean?). If published, this will include your full peer review and any attached files.

Reviewer #1: No

Reviewer #2: **Yes: **Timothy Dennis Verstynen

Reviewer #3: No
---

## [Decision Letter · Decision Letter 1]

15 Aug 2024

Dear Prof. Bogacz,

Thank you very much for submitting your manuscript "Reward Bases: A simple mechanism for adaptive acquisition of multiple reward types" for consideration at PLOS Computational Biology.

As with all papers reviewed by the journal, your manuscript was reviewed by members of the editorial board and by several independent reviewers. In light of the reviews (below this email), we would like to invite the resubmission of a revised version.  Specifically, we agree with Reviewer #2 that his lingering concerns are valid and need to be addressed more thoroughly.  Furthermore, additional discussions with editorial board members have raised the following issues, which should also be addressed:

-- The mechanism for signaling internal states to the DA circuitry requires some additional justification, as does the immediacy of the changes in behavior that result. 

-- On a related note, more care should be taken to accurately compare this work to ref. [45] and the capabilities of homeostatic RL.  

-- We call the authors attention to:  Duriez, Alexia, et al. "Homeostatic reinforcement theory accounts for sodium appetitive state-and taste-dependent dopamine responding." *Nutrients* 15.4 (2023): 1015.  This paper is also relevant to the authors' work and merits discussion in the context of the submitted manuscript.

We cannot make any decision about publication until we have seen the revised manuscript and your response to the reviewers' comments and to those above. Your revised manuscript is also likely to be sent to reviewers for further evaluation.

Sincerely,

Jonathan Rubin

Academic Editor

PLOS Computational Biology

Lyle Graham

Section Editor

PLOS Computational Biology

Reviewer's Responses to Questions

**Comments to the Authors:**

Reviewer #1: The authors have addressed all my concerns and I do not have any additional comments.

Reviewer #2: The authors have done a great job addressing most of my comments and concerns. The revised manuscript is substantially stronger. However, there appears to be a small conundrum surrounding my first point.

In the original review I pointed out that the models of the data on dopamine coding of errors in different dimensions (pg. 9) had an inferential error by violating the hierarchical principle. Specifically the model with a main effect of value and an interaction term for value and stimulus

Identity was selected as the “best” fit, even though the main effect term of stimulus identity was not included in the model. Indeed, as the authors point out in their reply that the correct form of this model (with both main effects along with the interaction term) provides a substantially worse fit to the data than the incorrect model (and even worse than the stimulus only model) (Fig. 4A). Suggesting that, in the space of these feature sets, the model they infer from is not, in fact, the best model.

The authors reply that this incorrect form of the model is equivalent to a later model including the main effect of food prediction error and the main effect of juice prediction error. I’ve spent the better part of today trying to work through that equivalence proof, but to no luck (admittedly I’m not a very good mathematician).

So I have few lingering concerns here:

Even though this model is later shown to be equivalent to a main effects only model, in the space of features being evaluated in this set of comparisons, it is not in fact a winning model because it is not a valid form of the model. The correct form of the model is indistinguishable (or worse than) a stimulus only model, making the inference from *these* feature comparisons ambiguous at best.

Could the authors provide more details on the equivalence proof for these two models? This is a critical inference being made to justify the reward bases model and if I’m getting stuck on the equivalence justification, I have a feeling many other readers might get stuck as well.

I get what the authors are trying to do here by building up to their model, but it seems really odd to have a feature test comparison where the model that “wins” the comparison tests is statistically ill-formed (i.e., violates the hierarchical principle), but later shown to be equivalent to a model that is properly formed (i.e., a simple main effect model with different features). The reader is asked to hold a contradiction in their head until later (or read very very carefully and work out a proof on their own). So now I am confused why the versions of the models run in this section (and shown in Figure 4) are relevant if the goal is to just get to the reward bases model. Why not just jump straight into a reward bases model?

Signed: Timothy Verstynen

Reviewer #3: The authors have comprehensively addressed the reviewers comments.

Congratulations on this interesting and detailed work.

**Have the authors made all data and (if applicable) computational code underlying the findings in their manuscript fully available?**

Reviewer #1: Yes

Reviewer #2: Yes

Reviewer #3: Yes

PLOS authors have the option to publish the peer review history of their article (what does this mean?). If published, this will include your full peer review and any attached files.

Reviewer #1: No

Reviewer #2: **Yes: **Timothy Verstynen

Reviewer #3: **Yes: **Nadav Amir
---

## [Decision Letter · Decision Letter 2]

22 Oct 2024

Dear Prof. Bogacz,

We are pleased to inform you that your manuscript 'Reward Bases: A simple mechanism for adaptive acquisition of multiple reward types' has been provisionally accepted for publication in PLOS Computational Biology.

Best regards,

Jonathan Rubin

Academic Editor

PLOS Computational Biology

Lyle Graham

Section Editor

PLOS Computational Biology

Reviewer's Responses to Questions

**Comments to the Authors:**

Reviewer #2: Outstanding job on addressing my lingering concern. Thank you for being patient with my suggested revisions.

-Tim Verstynen

**Have the authors made all data and (if applicable) computational code underlying the findings in their manuscript fully available?**

Reviewer #2: Yes

PLOS authors have the option to publish the peer review history of their article (what does this mean?). If published, this will include your full peer review and any attached files.

Reviewer #2: **Yes: **Timothy Verstynen

---

## [Editor Report · Acceptance letter]

7 Nov 2024

PCOMPBIOL-D-24-00693R2 

Reward Bases: A simple mechanism for adaptive acquisition of multiple reward types

Dear Dr Bogacz,

I am pleased to inform you that your manuscript has been formally accepted for publication in PLOS Computational Biology. Your manuscript is now with our production department and you will be notified of the publication date in due course.

With kind regards,

Anita Estes
